# TASKGALAXY:
# SCALING MULTI-MODAL INSTRUCTION FINE-TUNING WITH TENS OF THOUSANDS VISION TASK TYPES

**Jiankang Chen**[*]**, Tianke Zhang, Changyi Liu, Haojie Ding, Yaya Shi, Feng Cheng,**
**Huihui Xiao, Bin Wen**[†]**, Fan Yang, Tingting Gao, Di Zhang**
Kuaishou Technology
{chenjiankang03,zhangtianke,liuchangyi,wenbin,yangfan}@kuaishou.com

## ABSTRACT

Multimodal visual language models are gaining prominence in open-world applications, driven by advancements in model architectures, training techniques, and high-quality data. However, their performance is often limited by insufficient task-specific data, leading to poor generalization and biased outputs. Existing efforts to increase task diversity in fine-tuning datasets are hindered by the labor-intensive process of manual task labeling, which typically produces only a few hundred task types. To address this, we propose TaskGalaxy, a large-scale multimodal instruction fine-tuning dataset comprising 19,227 hierarchical task types and 413,648 samples. TaskGalaxy utilizes GPT-4o to enrich task diversity by expanding from a small set of manually defined tasks, with CLIP and GPT-4o filtering those that best match open-source images, and generating relevant question-answer pairs. Multiple models are employed to ensure sample quality. This automated process enhances both task diversity and data quality, reducing manual intervention. Incorporating TaskGalaxy into LLaVA-v1.5 and InternVL-Chat-v1.0 models shows substantial performance improvements across 16 benchmarks, demonstrating the critical importance of task diversity. TaskGalaxy is publicly released at https://github.com/Kwai-YuanQi/TaskGalaxy.

## 1 INTRODUCTION

Recent breakthroughs in artificial intelligence have been fueled with the development of a large number of large multimodal models (LMMs) (Liu et al., 2024b; Bai et al., 2023; Liu et al., 2024a; Zhang et al., 2024c). These models are typically composed of a pre-trained visual encoder (Radford et al., 2021), a pre-trained large language model (Touvron et al., 2023a), and a lightweight structure (Q-former for BLIP2 (Li et al., 2023c), two layers of MLP for LLaVA (Liu et al., 2024b), etc.) connecting the above two, which have been adopted in various domains such as image captioning, object detection, visual question answering and other related fields. How to improve the model's performance in various mission scenarios is of great importance for deploying such a model into an open-world system.

To enhance the performance of LMMs in specialized and general-purpose domains, mainstream approaches focus on three key areas: optimizing model architectures (Alayrac et al., 2022; Wang et al., 2023; Zhang et al., 2023a; Li et al., 2023c; Liu et al., 2024a), improving training strategies (Wang et al., 2024; Sun et al., 2023; Banerjee et al., 2024; Zhu et al., 2024), and constructing high-quality data (Xu et al., 2024; Chen et al., 2023b; Shi et al., 2024). While advances in model architectures and training strategies are crucial, our work focuses on addressing the critical challenges in the data domain. Current multimodal models typically undergo a biphasic training process: a pre-training phase with image-text pairs for visual-textual alignment, followed by a supervised fine-tuning (SFT) phase with command-format data to refine multimodal abilities. However, the diversity of tasks in the pre-training phase is limited, affecting the generalization ability of visual language models (Xu et al., 2023). Recent research (Hu et al., 2024; Li et al., 2024b; Shi et al., 2024; Xu et al., 2023; Yin et al., 2024; Liu et al., 2024b; Zhao et al., 2024) has focused on expanding task diversity in the

---

[*]This work was done while Jiankang Chen was interning at Kuaishou.
[†]Corresponding author.

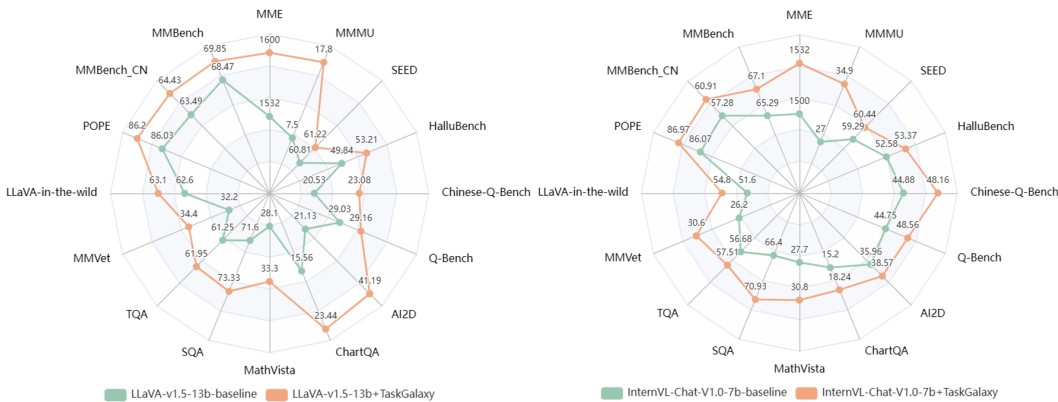

Figure 1: **An illustration of the benefits of high task type coverage in TaskGalaxy for the SFT stage.** We presented the performance of LLaVA-v1.5-13B and InternVL-Chat-v1.0-7B models, both before and after integrating TaskGalaxy into the fine-tuning dataset.

supervised fine-tuning phase to enhance instruction adherence and logical reasoning. Despite these efforts, instructional datasets still face limitations in task diversity, leading to performance bottlenecks. For example, Vision-Flan (Xu et al., 2024) proposed a comprehensive dataset with tasks like OCR and object detection but required extensive manual labeling and yielded only around 200 task types. Similarly, VisionLLM v2 (Wu et al., 2024) aggregated numerous task types but needed task-specific decoders, limiting the dataset's generalizability.

To address the limited task diversity in existing fine-tuning datasets and overcome current methodological constraints, we propose a novel pipeline for generating fine-tuning data. This pipeline enables the simultaneous construction of a wide variety of task types using contemporary multimodal models (Hurst et al., 2024; GLM et al., 2024; Chen et al., 2024b; 2023c). We introduce an innovative supervised fine-tuning dataset, **TaskGalaxy**, which encompasses over 19,000 hierarchical task types, significantly enhancing task diversity in multimodal scenarios. Specially, to minimize manual involvement and effectively construct multi-level multimodal task type data for fine-tuning dataset generation, our data generation pipeline is divided into five main steps as follows: (1) we initially designed several foundational multimodal graphic task types as seeds. These include tasks such as Optical Character Recognition (OCR) and Image Description for visual understanding, Logical Reasoning and Analysis for deep reasoning, as well as Multiple Choice Questions and Fill-in-the-Blank Questions for question-and-answer formats and so on. We then design appropriate prompts to enable GPT-4o to iteratively expand the range of task types both at the same level and to generate new, more detailed task types at subsequent levels. Notably, the number of task types per level is flexible and controllable, allowing for theoretically infinite generation. In our work, we have generated over 19,000 distinct types of tasks. (2) To provide image sources for all task types, we collected a diverse set of images from multiple open-source datasets. (3) We preliminarily select task types that are strongly correlated with each image using CLIP's (Radford et al., 2021) similarity between image embeddings and text embeddings. Appropriate prompt for GPT-4o is then designed to further filter task types corresponding to each image. (4) GPT-4o is utilized to generate question and answer pairs related to each task type for the selected images. (5) To ensure the reasonableness and alignment of the questions, task types, and images, we employ three robust open-source multimodal models GLM4v (GLM et al., 2024), InternVL2_26B (Chen et al., 2023c), and InternVL-Chat-v1.5 (Chen et al., 2024b) as judges for scoring and screening the generated pairs. Finally, we generate a novel high-quality fine-tuning dataset **TaskGalaxy** containing 19,227 task types and around 410k visual Q&A samples. After fine-tuning LLaVA-v1.5-7B & 13B and InternVL-Chat-v1.0-7B & 13B with the **TaskGalaxy** dataset along with raw fine-tuning data, the models showed an average improvement of 4.5 & 3.83, 3.0 & 3.64 points across fifteen benchmarks compared to the only original data fine-tuning, and Figure 1 illustrates the performance gains achieved by LLaVA-v1.5-13B and InternVL-Chat-v1.0-7B after incorporating TaskGalaxy during the SFT stage. Additionally, it demonstrated an increase of 68 points for LLaVA-v1.5-13B on the MME benchmark, indicating that our dataset enhances the model's generalization ability.

The contributions of this study are as follows.

- We propose a novel multi-modal instruction fine-tuning dataset, **TaskGalaxy**, which contains tens of thousands+ of vision task types and approximately 413k samples, addressing the limitation of task diversity in existing datasets.
- An almost fully automated pipeline for creating a comprehensive fine-tuning dataset of diverse task types was designed. This pipeline can be flexibly expanded by incorporating high-quality images, task types, and question-answer samples.
- Incorporating **TaskGalaxy** into the fine-tuning of LLaVA-v1.5 and InternVL-Chat-v1.0 resulted in improvements across all 16 benchmarks compared to fine-tuning with the original data which proves that expanding the diversity of visual task types and high-quality question-answer pairs associated with these tasks significantly enhances the generalization capabilities of multimodal models.

## 2 TASKGALAXY DATASET

### 2.1 OVERVIEW

The TaskGalaxy dataset consists of 19,227 hierarchical task types, ranging from OCR and image description to fine-grained object recognition and complex logical reasoning and so on. These task types originate from a small set of manually defined seeds, further expanded by GPT-4o through prompt design. CLIP's graphical similarity is used to filter task types that best match specific images. GPT-4o then generates corresponding question-answer pairs, which are evaluated by three open-source models to select the highest-quality samples. This process yields 413,648 high-quality question-answer pairs aligned with these hierarchical task types.

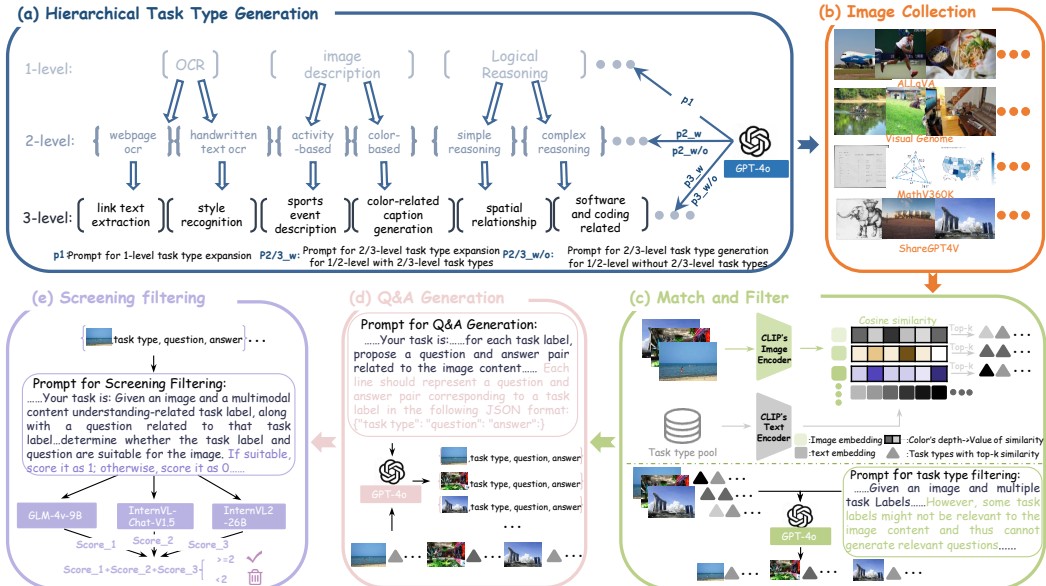

Figure 2: **An overview of the task type and high-quality question-answer pairs generation pipeline for TaskGalaxy**. We initially define the first level of visual task types, along with a small number of second and third level task types. Subsequently, we instruct GPT-4o to extend these to a broader range of task types. We then collect image modalities from existing publicly available datasets for matching task types with images, filtering, generating question answers related to task types, and utilizing the three referee models to obtain final high-quality visual quiz pairs for various task types strongly related to images.

### 2.2 DATASET GENERATION PIPELINE

In this section, we outline the pipeline for creating the **TaskGalaxy** dataset, highlighting the reduction of human intervention and leveraging multiple advanced LLMs. The process encompasses task

type expansion, precise matching between task types and images, automated question-answer generation, and referee-based filtering to ensure data quality, with the majority of these steps being fully automated. **TaskGalaxy** stands out from other fine-tuning datasets due to its extensive visual task diversity, minimal human involvement, rich and high-quality task-related Q&A data, and scalability. The workflow is depicted in Figure 2.

To ensure these key attributes of the **TaskGalaxy** dataset, we have designed a five-step pipeline:

**Prompt-task type Generation:**

You are an expert in multimodal content understanding with extensive experience in this field. I would like to construct a comprehensive task labeling system related to multimodal content understanding that includes only two modalities: images and text. In this system, the input consists of images and corresponding text, and the output is in the form of text. I want to start constructing this system from the first-level task categories. Currently, dozens of first-level task categories have already been established by humans, which are:{task_type} Please expand and supplement with new first-level categories that do not belong to the aforementioned categories to ensure comprehensive coverage of all task categories in multimodal content understanding scenarios. Output format: Each line should correspond to one category, without any other characters.

Figure 3: **The prompt template used in GPT-4o API for first-level task type generation.**

**Hierarchical Task Type Generation.** To address the challenge of expanding the range of visual task types in the multimodal domain, manual organization is impractical due to its time-intensive nature. Instead, we leverage a robust multimodal model to automate the construction of diverse task types with minimal human intervention. Rather than randomly generating task types, we designed a hierarchical method. As shown in Figure 2(a), we began by manually defining a small set of first-level task types, such as OCR, Image Description, Logical Reasoning. Secondary and tertiary task types, such as OCR~webpage OCR, Logical Reasoning~Complex Reasoning~software and coding, and Detection~Anomaly Detection, were also specified. These initial task types served as seeds, guiding GPT-4o to expand them at multiple hierarchical levels using specially designed prompts. The detailed prompt for first-level task type expansion is illustrated in Figure 3, and descriptions of the prompts for secondary and tertiary tasks are included in the Appendix. These prompts have been carefully designed to mitigate the risk of overlap between different task types. Specifically, as illustrated in Figure 3, the instruction "Please expand and supplement with new primary-level categories that do not belong to the aforementioned categories"( refer to Table A-2 in the Appendix for additional prompts that generate non-overlapping task types) ensures this objective. This process resulted in the generation of 19,227 hierarchical task types, making TaskGalaxy the most diverse dataset of its kind. Further details on these task types are provided in the Appendix.

Table 1: **Summary of Image Sources for the Collection.** We have curated a diverse array of image sources, ranging from natural images (Lin et al., 2014; Kirillov et al., 2023; Xu et al., 2024) to specialized (Mathew et al., 2022; Lindström & Abraham, 2022; Kafle et al., 2018; Seo et al., 2015; Chang et al., 2022; Lu et al., 2022b) and web-based images (Schuhmann et al., 2021; Ordonez et al., 2011; Sharma et al., 2018). This selection aims to ensure the diversity and comprehensiveness necessary to address various hierarchical task types effectively.

| Name | Image Source | Samples |
|---|---|---|
| ALLaVA | LAION, VFLAN | 326K |
| Visual Genome | MSCOCO | 108K |
| MathV360K | DocVQA, IconQA, UniGeo, CLEVR-Math, etc | 31K |
| ShareGPT4V | COCO, SA, SBU, etc | 358K |

**Image Collection.** To align with the diversity and comprehensiveness of the hierarchical task types, as illustrated in Figure 2(b), we collected approximately 800k images from a variety of sources, as shown in Table 1, including ALLaVA (Chen et al., 2024a), Visual Genome (Krishna et al., 2017), MathV360K (Shi et al., 2024), and ShareGPT4V (Chen et al., 2023b). These sources cover a range of tasks, such as object detection (COCO (Lin et al., 2014)), visual question answering (VFLAN (Xu et al., 2024)), segmentation (SA (Kirillov et al., 2023)), and document-related tasks

(DocVQA (Mathew et al., 2022)). Additionally, CLEVR-Math (Lindström & Abraham, 2022) contributes to mathematical problem-solving tasks, while LAION (Schuhmann et al., 2021) provides a broad array of web-sourced images, including artwork and watermarked content. All collected images maintain their original resolution. Additionally, to provide further insight into the image data, the sample size statistics for all image source datasets are provided in the Appendix. Our highly scalable data pipeline can accommodate diverse image types, requiring only a single modality, thus highlighting its flexibility and adaptability.

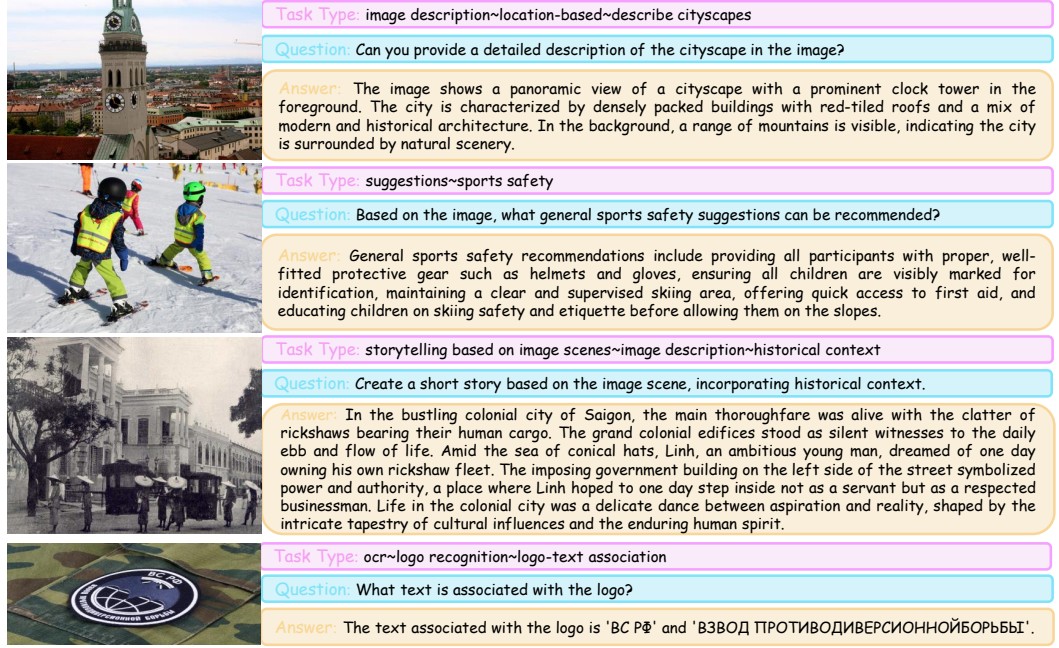

Figure 4: **Sample images, task types, and Q&A in TaskGalaxy**. The **Task Type** refers to the visual task related to the image. **Question** and **Answers** are generated by GPT-4o and subsequently filtered by three refereeing models.

**Match and Filter.** Given the complexity and labor-intensive nature of selecting images for a wide range of hierarchical task types, we utilize the vision language model CLIP (Radford et al., 2021), renowned for its robust text-to-image alignment capabilities, to streamline the initial screening process. CLIP is utilized to identify task type names that best correspond to each image, improving both efficiency and accuracy in matching. As illustrated in Figure 2(c), for any image $x_i$ and task type text $t_j$, we pass them through CLIP's image encoder $\mathcal{I}$ and text encoder $\mathcal{T}$ to obtain their respective embeddings. The match between the image and the task type is then calculated using the following formula:

$$s_j = \mathcal{I}(x_i) \cdot \mathcal{T}(t_j), \tag{1}$$

where $\cdot$ denotes the cosine similarity between image embedding and text embedding, $x_i$ from image pool, $t_j$ from task type pool. To identify the most appropriate task type for a given image, we initially select the $k$ task types that exhibit the highest degree of similarity and $k = 10$ in our pipeline. These selected task types form the preliminary sequence for that image.

Given that task types with the highest $k$ similarity have been initially assigned to each image, some mismatches may still persist. To further refine and filter the task types that are more relevant to the image, we use a specially designed prompt, detailed in the appendix, to guide GPT-4o to further analyze and filter the task types from the initial sequence, identifying those that most closely match the image. The resulting list of task types will serve as the final selection for subsequent processes. Examples of images and their corresponding task type lists are shown in Appendix.

**Q&A Generation.** A key factor in improving multimodal models for visual question-answering tasks is high-quality training data with diverse question-answer pairs. To address this, we developed

high-quality image-matched task types, as illustrated in Figure 2(d). We designed specific prompts, detailed in the appendix, to guide GPT-4o in generating questions and answers based on image content and task-type text. This approach ensures diverse question-answer pairs that cover a broad spectrum of visual Q&A scenarios. The resulting data provide a comprehensive training resource, enabling the model to learn from a wide range of scenarios and enhance its capabilities.

**Referee Screening.** Despite GPT-4o's superior performance, it may still encounter mismatches between task types, questions, and images. To improve alignment and manage the high costs of closed-source APIs, we used three high-performing open-source multimodal models (Chen et al., 2023c; 2024b; GLM et al., 2024) to evaluate task types and questions for each image, as shown in Figure 2(e). Following the principle that "three heads are better than one," these models assessed each task type and question relative to an image on a binary scale, awarding one point for a match and zero otherwise. We integrated these scores, selecting only those with a cumulative score of two or more for inclusion in the TaskGalaxy dataset. This process enhances the accuracy of task type, question, and image matching. To ensure balance across task types, we randomly selected 1-55 samples per task type from the final dataset.

Figure 5 illustrates the distribution of sample counts across different task types. To provide a more comprehensive understanding of TaskGalaxy, we analyze not only the proportion of samples in each task type but also the hierarchical distribution of task types across layers. Specifically, the ratio of task types in the first three layers follows a 1:2:3 pattern, corresponding to 115, 2796, and 14,370 task types, respectively. TaskGalaxy is a high-quality, supervised fine-tuning dataset covering a diverse range of tasks. Figure 4 presents sample images along with their associated task types, questions, and answers from TaskGalaxy.

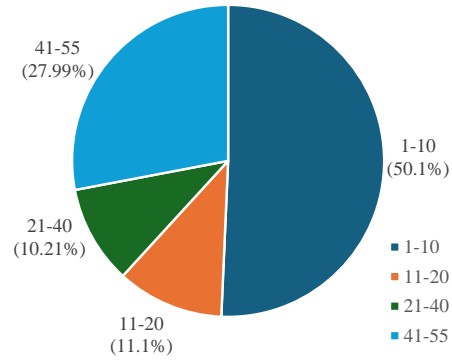

Figure 5: **Distribution of the number of images across the 19,227 task types in TaskGalaxy.** The ranges 1-10, 21-40 and etc. indicate the number of samples associated with different task types in TaskGalaxy. The corresponding ratios represent the proportion of task types that fall within each specified sample range.

Our task types and image data are designed to be expandable. The TaskGalaxy dataset can be automatically updated with additional images and task types through the fine-tuning data pipeline described earlier. This scalability facilitates future iterations, offering opportunities for further updates. To assess the characteristics of our dataset in relation to previous studies, we provide a comparative analysis between our dataset and previous work in the Appendix. And to comprehensively present properities of TaskGalaxy, a detailed dataset card is available at TaskGalaxy Data Card (PDF).

## 3 EXPERIMENT

### 3.1 EXPERIMENT SETUP

**Model Architecture.** In the matching and filtering stage, the CLIP-L/14Radford et al. (2021) model, developed by OpenAI, is employed. This model utilizes a ViT-L/14 Transformer architecture as the image encoder and a masked self-attention Transformer as the text encoder. In the stage of evaluating the TaskGalaxy dataset, We use the LLaVA (Liu et al., 2024b) and InternVL-Chat-v1.0 (Chen et al., 2023c) models. Both models feature a pre-trained visual encoder and a large language model, linked by a two-layer MLP projection layer. LLaVA employs a two-stage training: initially, a subset of the CC3M (Sharma et al., 2018) dataset pretrains only the projection layer for multimodal alignment. We use the model pre-trained in this phase as the basis for fine-tuning in the subsequent phase. For validation, we selected two variants: LLaVA-v1.5-7B and LLaVA-v1.5-13B. Similarly, InternVL-

Chat-v1.0 undergoes two training phases: first, the MLP layers are trained with the LGS-558K (Liu et al., 2024b) dataset, followed by training the language model with the LLaVA-Mix-665K (Liu et al., 2024a) dataset.

**TaskGalaxy Finetuning.** We incorporated our TaskGalaxy dataset with the original supervised fine-tuning data organized by LLaVA and are gathered from publicly available academic task-oriented datasets (Kazemzadeh et al., 2014; Krishna et al., 2017; Marino et al., 2019; Mishra et al., 2019) during the second stage of model training for both models. In this stage, the visual encoder weights were frozen, while the projection layer and large language model weights were fine-tuned.

**Benchmarks.** To design a comprehensive instruction fine-tuning dataset for various vision tasks, we selected 16 benchmarks to evaluate model performance. These include MME-Perception (MME) (Fu et al., 2023), which measures perception abilities across 14 subtasks; MMBench (MMB) (Liu et al., 2023), a multi-choice test covering all proficiency levels; MMBench_CN (MMB$^{CN}$) (Liu et al., 2023), the Chinese version of MMBench; POPE (Li et al., 2023e), which assesses illusion using subsets of COCO (Lin et al., 2014) (random, normal, adversarial); LLaVA-in-the-wild (LLaVA$^W$) (Liu et al., 2024b) and MMVet (Yu et al., 2023), evaluating visual conversation abilities and leveraging GPT-4o for response evaluation; TextVQA (TQA) (Singh et al., 2019), focused on text-related visual question answering; ScienceQA (SQA) (Lu et al., 2022a), with multiple-choice questions on scientific topics requiring visual answers; MathVista (Lu et al., 2023), assessing mathematical reasoning in visual contexts; ChartQA (Masry et al., 2022), evaluating visual and logical reasoning over charts; AI2D (Kembhavi et al., 2016), for diagram interpretation; Q-Bench (Wu et al., 2023), testing low-level visual perception and understanding; Chinese-Q-Bench (Q-Bench$^{CN}$), the Chinese version of Q-Bench; HallusionBench (HalluBench) (Guan et al., 2024), focusing on visual and knowledge illusions; SEED-Bench (SEED) (Li et al., 2023b), evaluating performance on both images and videos (with video accuracy assessed using sampled frames from LLaVA); and MMMU (Yue et al., 2024), which tests multimodal models on large-scale multidisciplinary tasks requiring advanced subject matter knowledge and reasoning skills.

**Implementation Details.** For LLaVA-v1.5, we utilize the pre-trained projection layer weights from LLaVA to fine-tune both the projection layer and the large language model (LLM). This fine-tuning is conducted using the 665k supervised fine-tuning data from LLaVA, supplemented by our TaskGalaxy dataset. We use two LLaVA architectures, Vicuna-13B v1.5 and Vicuna-7B v1.5 (Chiang et al., 2023), combined with CLIP-ViT-L-336px (Radford et al., 2021) and two layers of MLP as our visual-language models (VLMs). During instruction tuning, we fine-tune the MLP layer and LLM using 8 A800 GPUs, with an initial learning rate of 2e-5, a batch size of 16 per device, for 1 epoch, totaling approximately 9300 steps. For InternVL-Chat-v1.0, we first train the MLP layers with the LGS-558K dataset using 8 A800 GPUs, with an initial learning rate of 1e-3 and a batch size of 32 per device. Subsequently, we fine-tune the MLP layers and the LLM with both the raw fine-tuning dataset and the TaskGalaxy dataset, using the same hyperparameters as for LLaVA. We employ two InternVL-Chat-v1.0 architectures, Vicuna-7B and Vicuna-13B, along with InternViT-6B (Chen et al., 2023c).

## 3.2 Quantitative Comparison

Table 2 presents a quantitative comparison of LLaVA-v1.5 and InternVL-Chat-v1.0 models trained on the original fine-tuned data versus those fine-tuned with TaskGalaxy. The new models show improvements of 4.5 and 3.83 points across all 15 benchmarks for LLaVA-v1.5-7B and 13B, respectively, excluding MME. For InternVL-Chat-v1.0, the improvements are 3.0 and 3.64 points. It is worth noting that LLaVA-v1.5-13B sees a performance increase of 68 points with TaskGalaxy on the MME Benchmark.

Taking LLaVA-v1.5-7B as an example, we observed a 3.35% and 3.1% improvement over the original baseline on MMBench and MMBench_CN, respectively. For LLaVA-in-the-wild, we achieved a 3.3 points increase, demonstrating that the TaskGalaxy fine-tuning dataset enhances the model's performance in detailed description and complex reasoning tasks. Notably, incorporating TaskGalaxy resulted in improvements of 0.77, 1.75, 4.7, 5.48, and 12.94 points on TQA, SQA, MathVista, ChartQA, and AI2D, respectively, highlighting the dataset's broad coverage. On hallucination mitigation tasks, improvements of 1 to 2 points on POPE and HalluBench suggest that a diverse range of tasks helps address hallucination issues. Additionally, on the SEED benchmark, which includes

Table 2: **Elucidate the advantages of incorporating our TaskGalaxy dataset into SFT phase.** We present the performance comparison on existing representative benchmarks before and after incorporating TaskGalaxy. The results in the Baseline represent our re-implementation of the officially provided checkpoint. All the numbers are presented in % except MME and the full score is 100%. The indicator of MME is the perception score, the maximum value is 2000.

| Model | Method | Benchmarks | | | | | | | | |
|---|---|---|---|---|---|---|---|---|---|---|
| | | MME | MMB | MMB$^{CN}$ | POPE | LLaVA$^{W}$ | MMVet | TQA | SQA | MathVista |
| LLaVA-v1.5-7B | Baseline | 1506 | 64.69 | 58.07 | 85.9 | 53.0 | 25.9 | 58.21 | 69.51 | 26.7 |
| | Baseline+TaskGalaxy | **1533** | **68.04** | **61.17** | **86.7** | **56.3** | **29.9** | **58.98** | **71.26** | **31.4** |
| | | ChartQA | AI2D | Q-Bench | Q-Bench$^{CN}$ | HalluBench | SEED | MMMU | Average (w/o MME) | |
| | Baseline | 14.72 | 25.32 | 26.08 | 33.58 | 50.05 | 58.59 | 16.6 | 44.46 | |
| | Baseline+TaskGalaxy | **20.20** | **38.26** | **43.58** | **34.85** | **51.74** | **60.28** | **21.8** | **48.96** | |
| | | MME | MMB | MMB$^{CN}$ | POPE | LLaVA$^{W}$ | MMVet | TQA | SQA | MathVista |
| LLaVA-v1.5-13B | Baseline | 1532 | 68.47 | 63.49 | 86.03 | 62.6 | 32.2 | 61.25 | 71.60 | 28.1 |
| | Baseline+TaskGalaxy | **1600** | **69.85** | **64.43** | **86.20** | **63.1** | **34.4** | **61.95** | **73.33** | **33.3** |
| | | ChartQA | AI2D | Q-Bench | Q-Bench$^{CN}$ | HalluBench | SEED | MMMU | Average (w/o MME) | |
| | Baseline | 15.56 | 21.13 | 29.03 | 20.53 | 49.84 | 60.81 | 7.5 | 45.21 | |
| | Baseline+TaskGalaxy | **23.44** | **41.19** | **29.16** | **23.08** | **53.21** | **61.22** | **17.8** | **49.04** | |
| | | MME | MMB | MMB$^{CN}$ | POPE | LLaVA$^{W}$ | MMVet | TQA | SQA | MathVista |
| InternVL-Chat-v1.0-7B | Baseline | 1500 | 65.29 | 57.28 | 86.07 | 51.6 | 26.2 | 56.68 | 66.4 | 27.7 |
| | Baseline+TaskGalaxy | **1532** | **67.10** | **60.91** | **86.97** | **54.8** | **30.6** | **57.51** | **70.93** | **30.8** |
| | | ChartQA | AI2D | Q-Bench | Q-Bench$^{CN}$ | HalluBench | SEED | MMMU | Average (w/o MME) | |
| | Baseline | 15.20 | 35.96 | 44.75 | 44.88 | 52.58 | 59.29 | 27.0 | 47.79 | |
| | Baseline+TaskGalaxy | **18.24** | **38.57** | **48.56** | **48.16** | **53.37** | **60.44** | **34.9** | **50.79** | |
| | | MME | MMB | MMB$^{CN}$ | POPE | LLaVA$^{W}$ | MMVet | TQA | SQA | MathVista |
| InternVL-Chat-v1.0-13B | Baseline | 1525 | 65.64 | 60.31 | 86.00 | 51.7 | 28.7 | 56.94 | 70.12 | 28.7 |
| | Baseline+TaskGalaxy | **1534** | **69.50** | **63.14** | **86.43** | **52.9** | **32.0** | **59.51** | **72.72** | **30.5** |
| | | ChartQA | AI2D | Q-Bench | Q-Bench$^{CN}$ | HalluBench | SEED | MMMU | Average (w/o MME) | |
| | Baseline | 16.28 | 38.55 | 56.13 | 54.86 | 50.05 | 59.35 | 25.1 | 49.90 | |
| | Baseline+TaskGalaxy | **17.04** | **52.60** | **58.60** | **58.26** | **52.90** | **60.23** | **36.8** | **53.54** | |

12 evaluation latitudes and 19k questions, there was a modest 1.7 points improvement over the model trained solely on raw fine-tuned data. In low-level image evaluation tasks, Q-Bench and Chinese-Q-Bench, models fine-tuned with TaskGalaxy showed gains of 17.5 and 1.27 points, respectively. The improvements on the challenging MMMU benchmark, where the model showed a 5-point gain, are attributed to the inclusion of diverse task types, including math and humanities. We also observed a 27-point improvement on the MME benchmark. Across the 15 benchmarks excluding MME, TaskGalaxy led to an average improvement of nearly 4 points. Similar results were observed with the LLaVA-v1.5-13B, which achieved a remarkable 68-point gain on MME following TaskGalaxy fine-tuning. These improvements were consistent across InternVL-Chat-v1.0. To further demonstrate the effectiveness of TaskGalaxy, we provide additional comparative experiments which involve fine-tuning the aforementioned models, as well as more robust models, using TaskGalaxy alone and using other similar instruction-tuning datasets separately in the Appendix. The results are consistent with the findings presented earlier.

Our results demonstrate that TaskGalaxy, with its exceptionally broad coverage of task types, significantly enhances the performance of multimodal models across a wide range of tasks, offering valuable insights for the research community regarding task type diversity.

## 3.3 ABLATION STUDY

**The number of task types.** The primary objective of TaskGalaxy is to enhance the generalization capabilities of multi-modal models by encompassing a broad array of visual-language task types. TaskGalaxy includes a diverse set of 19,227 distinct task types. In this subsection, we examine how the number of task types affects the performance of multimodal models. We selected task types in increments of 2k, 3k, 5k, 10k, 15k, 18k, and 19,227 from TaskGalaxy, maintaining a constant total of 100k images, and conducted ablation experiments using the LLaVA-v1.5-7B model. As shown in Figure 6(left), benchmarks such as LLaVA-in-the-wild, ChartQA, AI2D, Q-Bench, and MMMU consistently improved with an increasing number of task types. The 'Average' performance across the 15 benchmarks, excluding MME, also shows a clear trend of enhancement with more task types, which is corroborated by MME performance changes in Figure 7. These results highlight the critical role of task type diversity in enhancing the capabilities of modern multimodal models.

**The number of samples.** In addition to the impact of the number of tasks, it is well established that the amount of sample data in the instruction fine-tuning dataset also significantly affects model performance. To investigate this, we conducted ablation experiments to assess the effect of varying data volumes on model performance. As depicted in Figure 6(right), we included all task types and

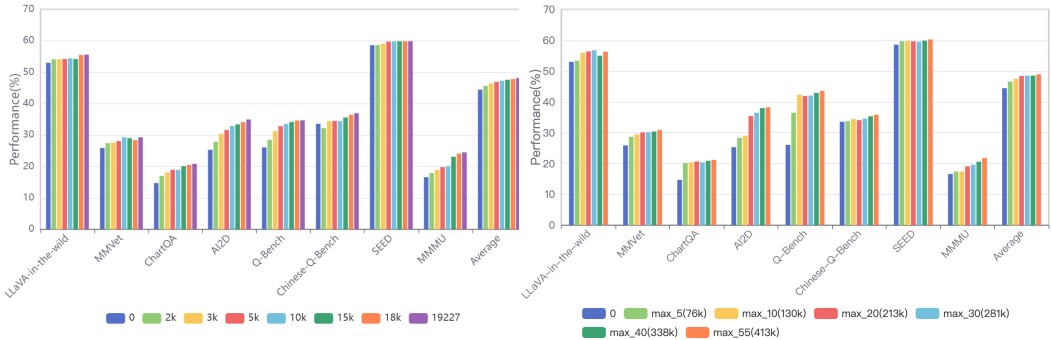

Figure 6: **Plot showing the change in baseline performance with variations in the number of tasks and the total number of samples.** Left: The effect of varying the number of task types (ranging from 2k to 19,227) on model performance, while maintaining a constant total sample size of 100k. Right: The impact on model performance of varying the number of samples across all task types, ranging from a maximum of 5 to 55 per task type (resulting in a total sample size from 76k to 413k), while keeping the number of tasks constant at 19,227.

controlled the variation in total sample size by setting the maximum number of samples for each task type, ranging from 5 up to 55 which corresponds to the final TaskGalaxy dataset. The results show that for benchmarks such as MMVeT, ChartQA, AI2D, and Q-Bench, as well as the average performance across 15 benchmarks excluding MME, performance generally improves with the increase in sample size. However, for LLaVA-in-the-wild, performance peaks at 281k samples, suggesting that the optimal sample size may vary depending on the specific benchmark. Nonetheless, there is a clear overall trend of increasing model performance with the increase in data volume.

## 4 RELATED WORK

**Large Multi-modal Models.** With the rapid advancement of large language models (LLMs), such as GPT-3 (Brown et al., 2020), LLama2 (Touvron et al., 2023b), InternLM (Team, 2023), and Baichuan 2 (Yang et al., 2023), there has been a growing focus on integrating visual knowledge into LLMs, exemplified by models like CLIP (Radford et al., 2021), BLIP (Li et al., 2022), and BLIP2 (Li et al., 2023c). While these models exhibit strong performance in graphic alignment and image captioning, they continue to face significant challenges in handling more complex visual question answering tasks. To enhance the model's instruction adherence and content understanding in visual question answering (VQA),

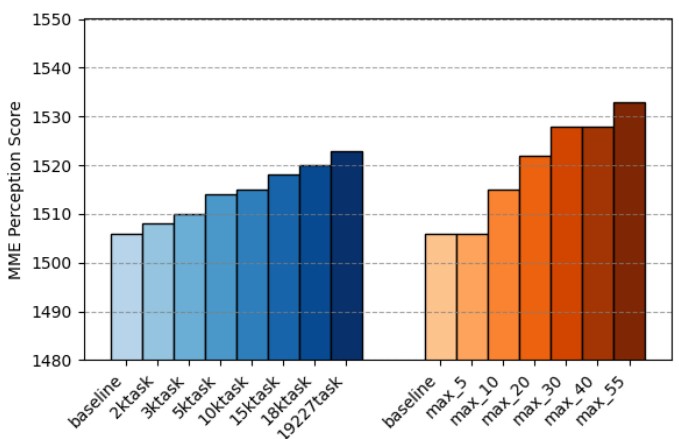

Figure 7: **Plot illustrating the variation in MME Perception performance scores in relation to the number of task types and the total number of samples.** The left bar represents performance across different numbers of task types, while the right bar represents performance across varying total sample sizes.

visual instruction fine-tuning strategies have garnered increasing attention in the training of large multi-modal models. For instance, models like LLaVA (Liu et al., 2024b), MiniGPT-4 (Zhu et al., 2023), and InstructBLIP (Dai et al., 2023) leverage large language models from the GPT4 (Achiam et al., 2023) family to generate fine-tuned instruction data, thereby enhancing performance in com-

plex VQA scenarios. Furthermore, to expand the range of VQA task scenarios, recent models such as LAMM (Yin et al., 2024) and MIMIC-IT (Li et al., 2023a), following the example of LLaVA, have extended their VQA capabilities to encompass 3D scenarios, multi-graph tasks, videos, and other complex domains. Recently, a series of open-source large multi-modal models with enhanced performance have consistently outperformed existing benchmarks. Notable examples include GLM-4v (GLM et al., 2024), Qwen-VL (Bai et al., 2023), InternLM-XComposer-2.5 (Zhang et al., 2024b), and InternLM2 (Zhang et al., 2024c), which are leading the field in various multimodal tasks. In addition to open-source models, recently developed closed-source models such as GPT-4v (OpenAI, 2023), GPT-4o (Hurst et al., 2024), and Claude-3.5 (Anthropic, 2024) continue to lead the field, often matching or surpassing open-source models in various VQA tasks. To facilitate comprehensive improvements in the performance of open-source models across a wide range of Visual Question Answering (VQA) tasks, we construct fine-tuning datasets featuring diverse task types based on multiple LMMs, while also leveraging closed-source models. This approach aims to enhance the performance of open-source models in various tasks.

**Multi-modal Instruction-Tuning Datasets.** Data-driven instruction fine-tuning strategies have become increasingly crucial in the training of multimodal models. Recent works have introduced high-quality instruction fine-tuning datasets designed to enhance models' visual question-answering capabilities. Among these datasets, MultiInstruct (Xu et al., 2023) is the first manually labeled multimodal instruction tuning benchmark dataset, encompassing 62 different multimodal tasks. Mini-GPT4 (Zhu et al., 2023) constructs an instruction-following dataset by combining image-text datasets with manually crafted instruction templates. LLaVA (Liu et al., 2024b) utilized the captions from the COCO dataset and the contents of the bounding boxes, sending them to GPT-4 to construct an instruction fine-tuning dataset comprising approximately 150k samples. Similar to LLaVA, LAMM (Yin et al., 2024) used the GPT API to generate command-response pairs for collected images and point clouds, resulting in approximately 190k graphic command-response pairs and 10k point cloud command-response pairs. However, these instruction fine-tuning datasets do not emphasize the concept of task types specific to VQA scenarios and lack diversity in the range of task types. Considering the concept of task types, VisionLLM v2 (Wu et al., 2024) aggregated hundreds of task types of multimodal data based on existing academic datasets but it requires the design of specific decoders for different tasks, which limits the generalizability of the dataset. Recent work VFLAN (Xu et al., 2024) enabled experts to construct 187 task types based on existing datasets, resulting in a large fine-tuning instruction dataset containing approximately 1,664k samples. However, this approach requires significant specialized manpower to annotate the extended tasks and generate the associated task structures, making it both time-consuming and labor-intensive. Additionally, despite the effort, the dataset covers only around 200 task types. In contrast, we developed TaskGalaxy, a high-quality instruction fine-tuning dataset, guided by the principles of maximizing the coverage of hierarchical task types for VQA while minimizing manpower investment. We successfully generated around 20,000 hierarchical task types and 410k VQA samples. Integrating TaskGalaxy into multimodal architectures like LLaVA and InternVL-Chat resulted in substantial performance improvements.

## 5 CONCLUSION

In this study, we present TaskGalaxy, a multi-modal instruction fine-tuning dataset comprising approximately 20,000 multi-modal task types and around 410k instruction Q&A samples. Additionally, we propose a pipeline for the systematic construction and generation of a diverse range of task types and corresponding high quality instruction Q&A samples. This approach addresses the limitations of existing multi-modal instruction fine-tuning datasets, particularly the narrow scope of task types and the excessive reliance on human intervention. TaskGalaxy encompasses an extensive range of multimodal visual Q&A tasks and offers a highly extensible pipeline that facilitates the addition of new task types and the generation of high-quality fine-tuned instructional data. We fine-tuned the LLaVA-v1.5 and InternVL-Chat-v1.0 models using TaskGalaxy, resulting in a significant improvement compared to using only raw fine-tunning data, respectively. Adequate empirical evaluation confirms the effectiveness of our broader task type data in enhancing the performance of multimodal models, highlighting the critical importance of task type diversity in the instruction fine-tuning dataset. We hope that our approach of constructing a dataset with a broad range of task types and reduced manual labor will guide future development of multi-modal instruction fine-tuning datasets and we plan to make the dataset publicly available for community research.

## 6 ETHICS STATEMENT

This study upholds rigorous ethical standards to ensure the credibility, confidentiality, and responsible usage of TaskGalaxy, a large-scale multimodal dataset designed for fine-tuning multimodal models. While the dataset does not directly involve human subjects, we developed it with careful attention to ethical considerations in data collection, processing, and dissemination. All data used in TaskGalaxy underwent thorough de-identification procedures to protect privacy and maintain anonymity. The images are sourced from publicly available, open-source datasets that comply with relevant copyright and usage laws, and we have clearly documented the licenses for these images. Additionally, we have provided a comprehensive license for TaskGalaxy itself, ensuring proper usage and distribution. The associated text is generated by GPT-4o, a model designed with safety guidelines to produce content that aligns with ethical standards. We maintained transparency and accountability throughout the dataset creation process, in full compliance with legal frameworks.

As TaskGalaxy is generated using automated models, we acknowledge the potential for biases in these models, such as GPT-4o and CLIP. To address these concerns, we employed multiple models and advanced filtering techniques, significantly mitigating the impact of these biases. Additionally, we encourage users to remain mindful of potential bias when applying the dataset in real-world scenarios. We are committed to continuously refining the dataset over time to further reduce bias and promote fairness. We have been transparent in documenting the creation and methodology of TaskGalaxy, sharing detailed information about the dataset's construction, filtering process, and quality control measures with the open-source community. This transparency allows for community feedback, collaborative improvements, and ensures the dataset is used in ways that align.

Throughout the development of TaskGalaxy, we adhered to rigorous ethical guidelines, prioritizing the rights and autonomy of contributors, and ensuring no academic misconduct occurred. We affirm our commitment to upholding ethical research practices and addressing any unintentional errors or oversights promptly. By releasing TaskGalaxy, we aim to advance research in multimodal learning while carefully considering its potential ethical and social impact, encouraging the academic community to use the dataset responsibly and be aware of the broader consequences in society.

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

# A  APPENDIX

## A.1  OVERALL REVIEW OF IMAGE SOURCES

Considering the accessibility of data sources and the task-related nature of the image data we aim to mine, we have opted for open-source image data. The approximate data sources and their corresponding sample sizes are presented in Table 1 of the main text. To provide further insight into the image data, the Table A-1 presents the statistics of the sample sizes for the different data sources collected.

Table A-1: Statistics of the sample sizes for the different data sources collected.

| Dataset | Sample Size | Dataset | Sample Size |
|---|---|---|---|
| UniGeo | 1,507 | GeoQA+ | 2,146 |
| GEOS | 64 | VizWiz | 749 |
| CLEVR-Math | 590 | PlotQA | 612 |
| VAQ2.0 | 2,327 | TQA | 1,137 |
| DocVQA | 2,672 | FigureQA | 1,970 |
| Geometry3K | 1,182 | MapQA | 590 |
| VQA-AS | 658 | A-OKVQA | 3,391 |
| VQA-RAD | 242 | PMC-VQA | 4,208 |
| TabMWP | 2,538 | Super-CLEVR | 971 |
| IconQA | 2,560 | DVQA | 1,660 |
| allava_laion | 145,359 | allava_vflan | 181,393 |
| Visual Genome | 108,249 | wikiart_images | 62 |
| ocr_vqa_images | 12,614 | coco_train2017 | 118,326 |
| web-celebrity_images | 67 | web-landmark_images | 57 |
| text_vqa_train_images | 3,485 | sam_images | 223,720 |
| share_textvqa_images | 55 | | |

## A.2  MORE DETAILS ON THE GENERATION PIPELINE PROCESS OF TASKGALAXY

**About Prompt:** In the TaskGalaxy dataset pipeline, the first step involves using GPT-4o to continuously expand new task types based on a set of human-defined task type seeds. This process requires designing distinct prompts for different levels of task types and determining whether lower-level task types exist within each hierarchy, allowing GPT-4o to systematically expand and populate the dataset. For level-1 task types, we focus solely on extending the existing prompt, as detailed in the main text. For generating two-level task types, the approach varies depending on whether the one-leveltask types have corresponding two-level tasks; prompts are designed accordingly to either continue expansion or generate new prompts. The same methodology applies to three-leval tasks. The detailed prompts are provided in Table A-2.

After generating a large number of hierarchical task types and collecting a substantial amount of open-source image data, the third step in our pipeline involves matching and filtering. Following the image-text cosine similarity matching conducted by CLIP, we proceed to further refine the selection of task types that are most compatible with specific images. In this stage, we employ a specially designed prompt for GPT-4o, denoted as p_filter in Table A-3, to filter and select the task types that best match each particular image.

After completing the matching and filtering of task types and images in the third stage, the fourth stage involves generating question-answer pairs related to all task types matched with the images. The prompt templates guiding GPT-4o to generate these task-type-related answer texts are denoted as p_Q&A in Table A-3.

After generating all the Q&A samples related to the task types, to further refine the selection of task types and questions that best match the images and ensure higher quality, we employ three open-source models in the final stage. This step focuses on filtering the images, task types, and their corresponding questions to identify the most suitable and coherent matches, while also considering

cost-effectiveness. The prompt templates used for this filtering process by the three open-source models are listed as p_openfilter in Table A-3.

In addition to the above prompt templates, in the ablation experiments, we aimed to verify the effectiveness of Chain-of-Thought (CoT) under multi-task type conditions. For a dataset with a total size of 76k, we prompted GPT-4o to articulate the reasoning process, guiding it to generate CoT answers. The prompt templates used for this purpose are denoted as p_CoT in Table A-3.

Table A-2: Different prompt templates were generated for the layered task types in the first stage. Here, p2_w denotes the continuation of expansion for a one-level task type that includes a two-level task, while p2_w/o represents the direct generation of a two-level task for a one-level task type without an existing two-level task. Similarly, p3_w indicates the continuation of expansion for a two-level task that includes a three-level task, and p3_w/o represents the direct generation of a three-level task for a two-level task type without an existing three-level task. The placeholder {...} denotes the corresponding task type field to be populated.

| Type | Prompt |
|---|---|
| p2_w | You are an expert in multimodal content understanding with extensive experience in this field. I want to construct a comprehensive task label system related to multimodal content understanding, which only includes image and text modalities. In this system, the input is an image and the corresponding text, and the output is in the form of text. The primary and secondary categories are connected by '~'. The task name of the primary category that needs to be detailed currently is {....}, and multiple secondary categories have already been established manually for this primary category, which are: {...}. Please supplement other categories that do not belong to the aforementioned secondary categories to cover all task scenarios under the primary category of multi-modal content understanding. Output format: Each line corresponds to one task category, without any other characters, and different levels of task categories are connected by '~'. |
| p2_w/o | You are an expert in multimodal content understanding with extensive experience in this field. I want to construct a comprehensive task label system related to multimodal content understanding, which only includes image and text modalities. In this system, the input is an image and the corresponding text, and the output is in the form of text. The primary and secondary categories are connected by '~'. The task name of the primary category that needs to be detailed currently is {...}. Please expand the secondary task categories under this primary task category to cover all tasks included in this primary task category in the context of multimodal content understanding. Output format: Each line corresponds to one task category, without any other characters, and different levels of task categories are connected by '~'. |
| p3_w | You are an expert in multimodal content understanding with extensive experience in this field. I want to construct a comprehensive task taxonomy for multimodal content understanding that includes only two modalities: images and text. The input to this taxonomy will be images and their corresponding text, and the output will be in text form. The primary, secondary, and tertiary categories in this taxonomy are connected by '~'. Currently, the task name for the secondary category that needs to be detailed is {...}, and several tertiary categories have already been established manually for this secondary category, which are: {...}. Please supplement additional categories that do not belong to the aforementioned tertiary categories to cover all tasks under the secondary category in the context of multimodal content understanding. Output format: Each line should correspond to one task category, without any other characters. Different levels of task categories should be connected by '~'. |
| p3_w/o | You are an expert in multimodal content understanding with extensive experience in this field. I want to construct a comprehensive task taxonomy for multimodal content understanding that includes only two modalities: images and text. The input to this taxonomy will be images and their corresponding text, and the output will be in text form. The primary, secondary, and tertiary categories in this taxonomy are connected by '~'. Currently, the task name for the secondary category that needs to be detailed is {...}. Please expand the tertiary task categories under this secondary task category to cover all tasks included in this secondary task category in the context of multimodal content understanding. Output format: Each line should correspond to one task category, without any other characters. Different levels of task categories should be connected by '~'. |

**About task types corresponding to the image:** In the third step of the data generation pipeline, the top 10 task types with the highest CLIP's image-text cosine similarity are assigned to all images. These task types are then further refined using GPT-4o to identify multiple task types corresponding to each image, which are subsequently used to generate question-answer pairs. Figure A-1 illustrates the list of images along with their associated task types.

Table A-3: The prompt templates used throughout the other stages of the TaskGalaxy pipeline: further filtered in the third stage based on image and task type compatibility, employed in the fourth stage to generate question-answer pairs, and finally refined in the fifth stage through multi-referee scoring to ensure optimal matching and high-quality alignment between task types, image content, and question-answer pairs, and the prompt template for Chain-of-Thought (CoT) answers generation in the ablation study.

| Type | Prompt |
|---|---|
| p_filter | You are a multimodal content understanding expert. Given an image and multiple task labels related to multimodal content understanding, with task labels as: init_task_type. I would like to generate some question-answer pairs related to these task labels based on the image content. However, some task labels might not be relevant to the image content and thus cannot generate relevant questions. Please fully understand the image content and the meanings of the task labels, and select all the task labels that are appropriate for this task. Ensure the task labels are the same as the original. The output format should be: [task labels], without any other characters. If there are no matches, output [None]. |
| p_Q&A | You are a multi-modal content understanding expert, very skilled in handling visual question answering tasks. Your task is: Given an image and task labels that are highly relevant to the content of this image, the task labels are: {...}. Please fully understand the content of the image and, for each task label, propose a question and answer pair related to the image content. Please try to propose some complex questions and provide answers to these questions. Please strictly follow the JSON format for the output. Each line should represent a question and answer pair corresponding to a task label in the following JSON format: {"task_type": "question": "answer":}. |
| p_filter_ referee | You are an expert in multimodal content understanding, particularly skilled in handling visual question answering tasks. Your task is: Given an image and a multimodal content understanding-related task label, along with a question related to that task label, the task label is "{task_type}", the question is "{question}", please fully understand the image content, the task label, and the question, and determine whether the task label and question are suitable for the image. If suitable, score it as 1; otherwise, score it as 0. Please only output your final score without any other characters. |
| p_CoT | You are a multimodal content understanding expert and you are very good at solving visual question answering tasks, I will give you an image and a question related to this image, the question is: {question}, your task is to fully understand the content of the image along with the question, please fully think about how the question should be answered, please give a step by step thought process of how you solved the question and finally output the answer to the question. |

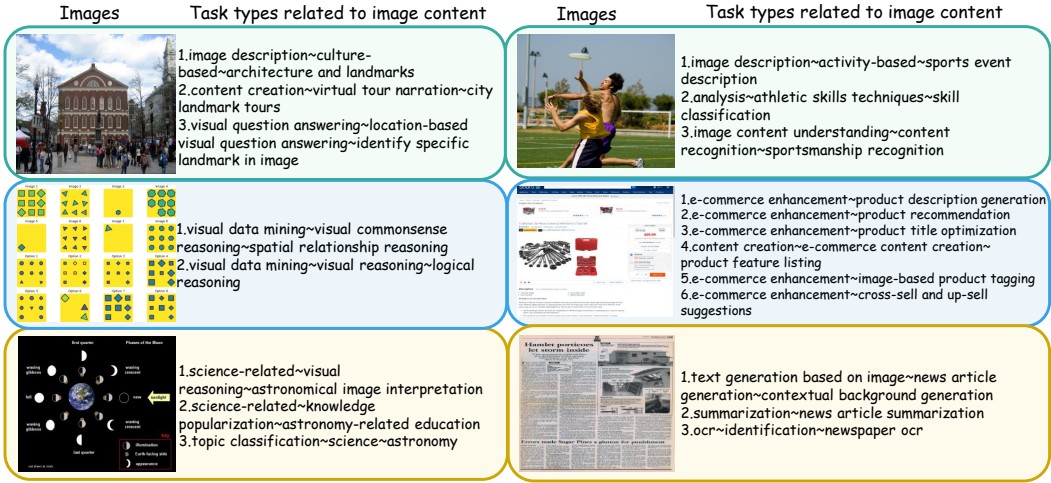

Figure A-1: Examples of images along with their corresponding task list pairs filtered by GPT-4o.

## A.3 COMPARISION WITH EXISTING DATASETS

Table A-4 presents a comparision between existing visual instruction tunning datasets and TaskGalaxy. For existing visual instruction tunning datasets, we directly adopt the numbers of tasks and instances reported in their original papers. VL-Qwen Bai et al. (2023) is a newly introduced large-scale dataset with human annotations; however, it is not publicly accessible. In contrast, MultiInstruct Xu et al. (2023) is constructed from publicly available datasets but primarily emphasizes visual grounding tasks, containing only 29 tasks that exclude region-specific information. To

further expand the number of task types, the recent Vision-Flan constructed a dataset comprising 196 task types through expert collection and curation. Although this approach increases the number of task types compared to previous datasets, it heavily relies on expert involvement, making the creation process highly time-consuming, labor-intensive, and significantly limiting its scalability. In contrast, TaskGalaxy employs an almost fully automated process to construct a high-quality instruction-tuning dataset encompassing nearly 20,000 task types—hundreds of times more than previous efforts.

Table A-4: Comparison between TaskGalaxy and existing visual instruction tuning datasets.

| Dataset | Instances | # Tasks | # Source |
|---|---|---|---|
| LLaVA Liu et al. (2024b) | 150K | 3 | Synthetic |
| LAMM Yin et al. (2024) | 196K | 8 | Synthetic |
| VL-Qwen Bai et al. (2023) | 350K | Unknown | Synthetic |
| M3IT Li et al. (2023d) | 2.4M | 40 | Private |
| mPlug-Owl Ye et al. (2023) | 150K | 3 | Synthetic |
| Shikra Chen et al. (2023a) | 156K | 4 | Synthetic |
| SVIT Zhao et al. (2023) | 4.2M | 4 | Synthetic |
| MultiInstruct Xu et al. (2023) | 510K | 62 | Synthetic |
| VISION-FLAN Xu et al. (2024) | 1.6M | 196 | Public |
| TaskGalaxy (Ours) | 431K | 19227 | Public+Synthetic |

## A.4 ILLUSTRATIONS AND ANALYSIS OF SAMPLES FILTERED DURING THE DATA GENERATION PIPELINE

In TaskGalaxy generation pipeline, there are two parts involved in matching and screening, and in the first part, we employ a two-phase process for matching and screening task types and images. In the first phase, task types are generated and open-source image data is collected. We use CLIP to perform an initial screening, matching task types with images based on their textual and visual similarity. However, the performance of CLIP's image-text matching is inherently limited. This sometimes leads to overestimation of similarity scores, resulting in mismatches where task types are paired with images that do not accurately represent their content. To address this limitation, the second phase involves leveraging GPT-4o with carefully designed prompts to refine the matches. This step effectively filters out task types that are not contextually related to the content of the images. Below, we provide a comparison illustrating the image-task pairings before and after the second-phase refinement by GPT-4o, demonstrating the improvement in alignment between task types and image content.

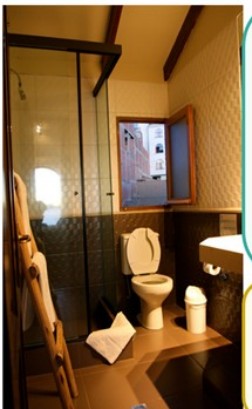

Figure A-2: Example 1 of Filtered-Out Samples.

In Figure A-2 depicting the bathroom of this hotel, the initial CLIP matching may generate terms like "public facilities", "restaurants", "smart home", "workplaces", and "museums" which are not

well-aligned with the actual image content. However, after applying corrective filtering with GPT-4o, the results are more compatible with the image content. This filtering refines the matches to include task types such as detailed descriptions of the room, hygiene-related information, tourist accommodation, and accommodation reviews, all of which align well with our expectations.

In Figure A-3 featuring the iconic building, the initial CLIP matching generated numerous task types related to communities, which were clearly not aligned with the content of the image. However, after GPT-4o filtering, only relevant task types were retained, such as those related to landmarks and weather background, ensuring a better match with the image content.

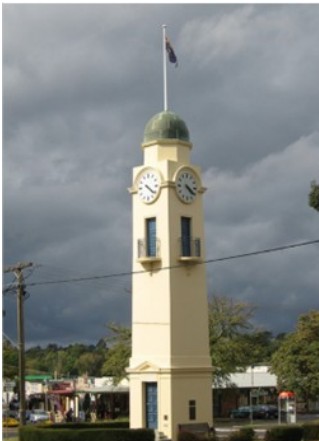

Figure A-3: Example 2 of Filtered-Out Samples.

In Figure A-4 showing the baseball sports, the initial CLIP matching generated numerous task types related to social media analysis. This may be influenced by the coexistence of social media content and sports within CLIP's training data. However, the images are more specifically aligned with the theme of sports. After GPT-4o filtering, the retained task types focus on sports recognition, player analysis, and other topics more relevant to the image content, providing a better contextual match.

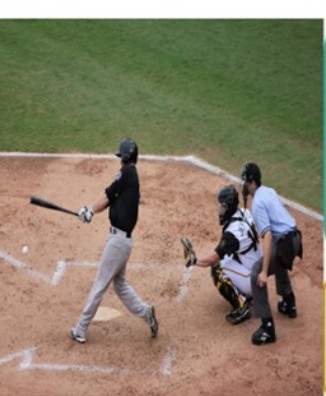

Figure A-4: Example 3 of Filtered-Out Samples.

In Figure A-5 showcasing restaurant food, the initial CLIP matching assigned task types such as restaurant OCR and related information. This could be because restaurant food is often associated with menus in the training data. However, after GPT-4o filtering, only food-related tasks, such as food recognition and other more relevant task types, were retained, ensuring a closer alignment with the image content.

**Analysis:** In the third part of the data generation pipeline, we utilize CLIP to initially match task types for each image. The primary purpose of using CLIP is to identify and screen the ten images

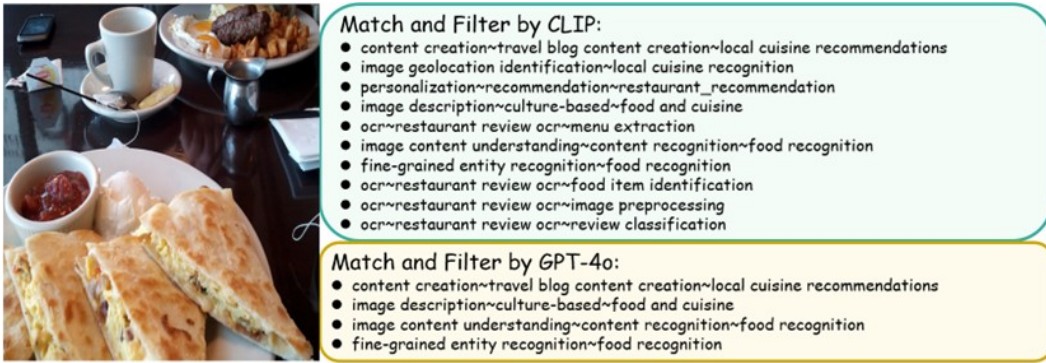

Figure A-5: Example 4 of Filtered-Out Samples.

most similar to the content of a given image. Additionally, this step helps avoid excessively long text prompts, which could exceed the input token limit during the subsequent GPT-4o screening stage. However, the effectiveness of the initial matching is influenced by the training data of CLIP, which can lead to the generation of illusory task types. For instance, during CLIP's training, food images may often co-occur with menus, sports images with media reports, natural scenery with tourism-related content, and pet images with pet products. These associations can cause CLIP to inaccurately match task types that are not directly relevant to the actual content of the image, resulting in imperfect matching in this first stage. To address these issues, the second stage involves GPT-4o, which refines the candidate task type list based on the actual content of the image. As demonstrated in the examples above, the filtered results effectively retain task types strongly aligned with the image content. This two-stage process ensures a higher degree of relevance and accuracy, achieving the intended purpose of the pipeline.

In the final step of the data generation pipeline, task-type-related question-answer pairs generated by GPT-4o are evaluated and screened using three open-source multimodal models. Samples are rejected in this step if at least two of the three models determine that the task type, questions, and answers are not well-aligned. The primary objective of this process is to eliminate low-quality samples that may arise from GPT-4o's hallucinations or incorrect responses. Below, we present several examples of samples that received a total score of less than 2 during the final scoring by the open-source models. These examples highlight cases where the task type, question, and answer alignment failed to meet quality expectations.

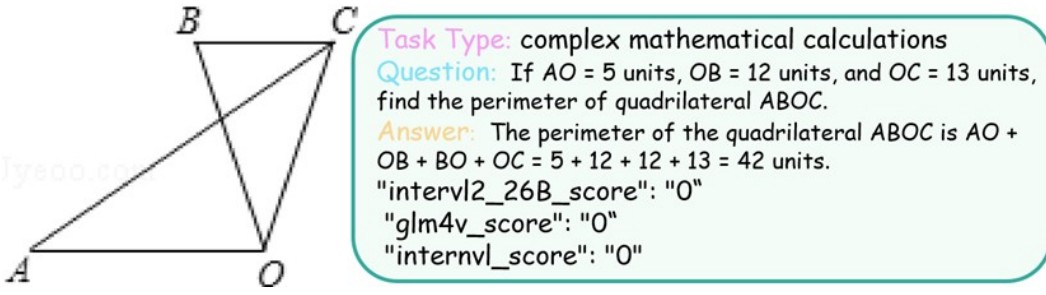

Figure A-6: Example 5 of Filtered-Out Samples.

For the geometric content-related questions in Figure A-6, the task type is categorized as complex mathematical operations. However, based on the task type and the picture content, GPT-4o generates unreasonable values for the lengths of OA, OB, and OC in the question. Furthermore, the calculated perimeter in the answer is incorrect. As a result, the MLLM (Multimodal Language Model) both formulates and answers the question incorrectly. Open-source models uniformly scored these samples as 0 points, leading to their exclusion.

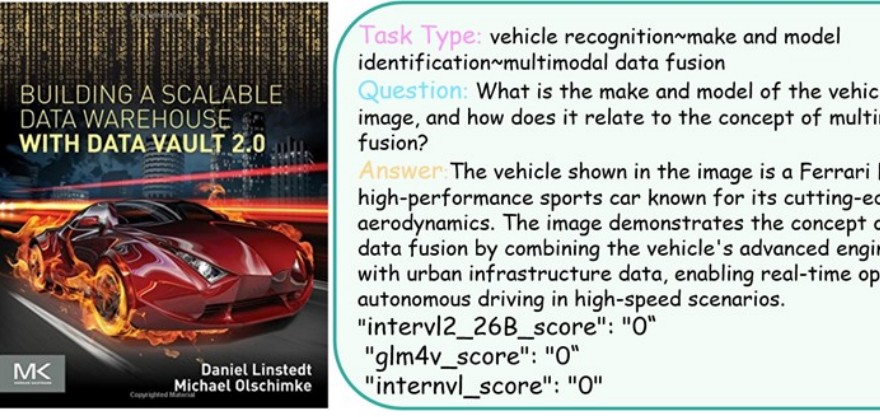

Figure A-7: Example 6 of Filtered-Out Samples.

The Figure A-7 features an artistically designed virtual vehicle, not a specific real-world brand. The response incorrectly identifies the vehicle as a Ferrari F8 Tributo. Additionally, there is no direct information about multimodal fusion embedded within the image. This aspect of the response is entirely speculative and fictional, likely influenced by the task type "data fusion," which has led to associating unrelated information with the image. The response attempts to connect keywords from the task type (e.g., multimodal data fusion) to content that is absent in the image. As a result, it diverges from the core task of identifying the vehicle's make and model. Instead, it focuses on explaining the concept of "multimodal data fusion," producing an answer that is misaligned with the task requirements. Due to this disconnection, the response was filtered out by the open-source model.

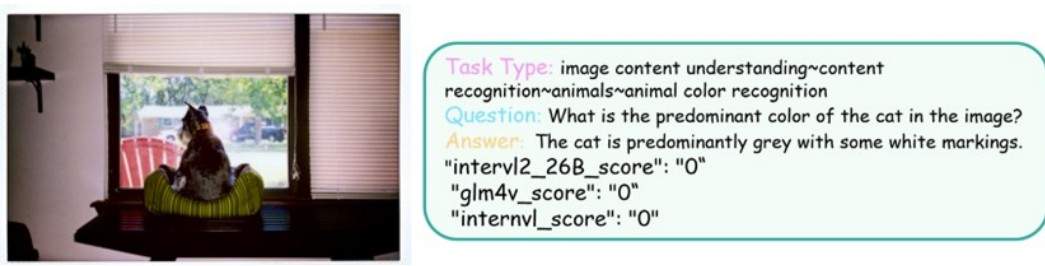

Figure A-8: Example 7 of Filtered-Out Samples.

In Figure A-8, the answer states that "The cat is predominantly gray with some white markings," but based on the content of the image, the animal in the picture is actually a dog sitting by the window, not a cat. This represents a classic case of hallucination, where the model incorrectly categorizes the animal as a cat. Additionally, the answer describes the animal as "mainly gray," whereas the image clearly shows the dog as predominantly dark brown or black, not gray. The task was to identify the 'predominant color' of the animal, but the mention of 'white markings' in the answer is not supported by the image content. This indicates that the model likely introduced assumptions or biases unrelated to the image itself when generating the answer, resulting in a response that does not align with the factual visual content. The open-source model received a score of 0 points due to its failure to correctly identify the animal species (mistakenly identifying a dog as a cat), its incorrect description of the animal's predominant color, and its generation of an answer based entirely on hallucinations unrelated to the factual content of the image.

In Figure A-9, the answer avoids addressing the core of the question, which concerns "the former owners of the painting and whether it is part of a prestigious collection." Instead, it merely states that detailed research into art records and auction histories is needed, failing to provide specific information directly related to the image content. Furthermore, the response entirely disregards key features of the image, such as the figure depicted and the style of the painting. It made no attempt to

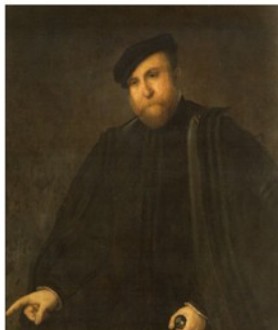

Figure A-9: Example 8 of Filtered-Out Samples.

analyze or extract relevant information from the image itself. The task type, "provenance tracking," explicitly requires specific details regarding the painting's provenance. However, the response deviates significantly from the task's objective by offering a generalized and non-informative statement, thereby failing to meet the expectations of the task. The open-source model received a score of 0 points due to its failure to extract useful information, its lack of analysis of the image content, and its inability to fulfill the task objectives.

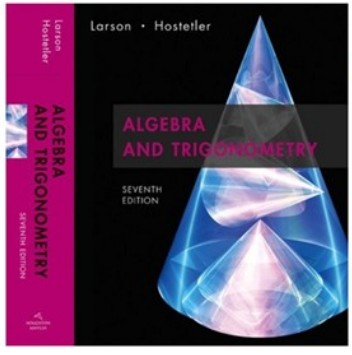

Figure A-10: Example 9 of Filtered-Out Samples.

In Figure A-10, the answer provided $z = \frac{5}{3}$ as the final solution, but through logical reasoning and algebraic calculations, $z$ can be correctly determined as 222 or $-\frac{5}{4}$. Thus, the answer is incorrect. The task requires logical reasoning and complex algebraic calculations to solve the problem, but the answer does not meet the expected level of complexity required for this task type. Additionally, the image content appears to be more relevant to OCR recognition of the book cover rather than the mathematical problem presented. This mismatch indicates that the problem itself may be irrelevant or disconnected from the graphical content, making the task type unsuitable. As a result, two of the three open-source models correctly judged the answer with 0 points. The Internvl model, however, provided a score, which may reflect a lack of mathematical reasoning capabilities, leading to an incorrect evaluation of the task's requirements and performance.

In Figure A-11, the model depicts a scenario in which "flooding inundates the marina and affects nearby buildings and facilities". The image shows no sign of flooding at all, only a normal marina scene with moored boats. The model's responses were generated entirely on the basis of fictional content. The model may have assumed flood damage in the image by default due to the task type "damage assessment" and ignored the normal conditions of the actual image. The task asks for an assessment of "infrastructure damage due to flooding" and there is no flooding in the image, so the model should recognize this and answer with something realistic such as "no flooding or damage observed". This mismatch between the task type and the image content led to the creation of subsequent illusory responses, which were filtered with a total score of less than two.

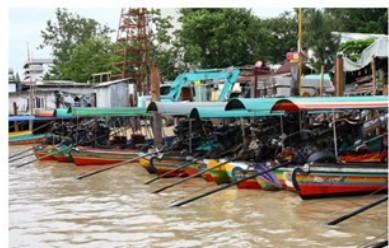

Figure A-11: Example 10 of Filtered-Out Samples.

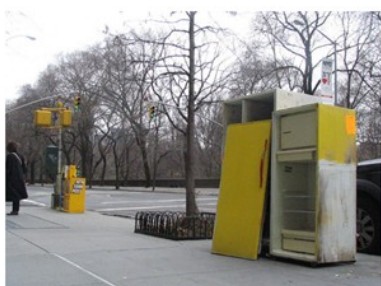

Figure A-12: Example 11 of Filtered-Out Samples.

In Figure A-12, the yellow structure in the picture appears to be a discarded refrigerator, and there is no visible notice board attached to it. The model likely generated a plausible but fabricated response based on the cues "yellow structure" and "notice board." However, the information in the response is entirely fictitious. Moreover, the yellow structure on the left clearly displays the text "New York Post," indicating it is a box for distributing newspapers. This has no connection to recycling dates or garbage sorting. The model incorrectly associated the task type (public message board) with garbage sorting information, likely due to a lack of OCR capability and proper semantic understanding, resulting in a response that is detached from the actual content of the image. Two open-source models correctly scored the response 0 points, recognizing that the content was illusory and inaccurate. However, one model mistakenly scored the response 1 point, possibly because the answer aligned loosely with the task type in a superficial manner.

**Analysis:** In the final stage of open-source model scoring and judging to screen samples, an analysis and summary of the above examples reveal several key issues. Some samples are unqualified due to errors in the questions or answers generated by the MLLM (Multimodal Large Language Model) itself. Others fail because the model's answers contain hallucinations or incorrect information, while in some cases, the task type does not align with the content of the image, resulting in the model's question-answer pair being invalid. Additionally, certain issues arise when the model generates hallucinatory information that, while related to the task type, is not connected to the actual content of the image. Moreover, the model's inherent capability limitations (e.g., OCR capability, mathematical reasoning ability) also contribute to errors in the answers. These challenges highlight the problems faced by MLLMs when generating answers. To mitigate the inclusion of such hallucinatory and erroneous samples in the final dataset, we incorporated three open-source MLLMs to evaluate and score samples. This multi-model evaluation approach helps identify unqualified samples more effectively. However, as observed in the examples above, even open-source models can sometimes misjudge samples. To address this, we adhere to the principle of "three ignorant cobblers working together outdo a Zhuge Liang," allowing multiple models to evaluate and score samples collectively. This strategy maximizes the chances of filtering out unqualified samples and ensures the construction of a high-quality TaskGalaxy dataset.

## A.5 LICENCES OF DATA

The licensing information for the image sources listed in Table 1 is as follows: ALLaVA (Apache License 2.0), Visual Genome (CC BY 4.0), MathV360K (Apache License 2.0), and ShareGPT4V (CC BY-NC 4.0). The proposed dataset, upon its open-source release, will be licensed under CC BY-NC 4.0.

We acknowledge our responsibility for ensuring legal compliance in data usage. The dataset licenses have been carefully reviewed, and our release under CC BY-NC 4.0 aligns with the restrictions of certain sources. Steps have been taken to mitigate potential legal risks and ensure adherence to the respective terms.

## A.6 EVALUATION REGULATIONS

For the evaluated benchmarks MME, MMBench, MMBench_CN, MM-VeT, POPE, SEED, SQA, and TextVQA, we utilized the official evaluation code provided by LLaVA. For AI2D, ChartQA, HallusionBench, LLaVA-in-the-wild, MMMU, Q-Bench, and Chinese-Q-Bench, we referred to the evaluation code that follows the official evaluation protocol of InternLM-XComposer. The Math-Vista evaluation baseline was conducted using the official evaluation code of MathVista. In benchmarks such as MM-VeT, LLaVA-in-the-wild, and MathVista, we replaced the original GPT-4 API with GPT-4o, which offers more stringent criteria and improved performance for scoring, answer extraction, answer matching, and related tasks.

## A.7 EXPERIMENTAL EVALUATION USING TASKGALAXY, BASELINE DATASET, AND OTHER INSTRUCTION-TUNING DATASETS INDIVIDUALLY

To demonstrate the impact of TaskGalaxy's task diversity on model performance, we compare the baseline fine-tuning data with TaskGalaxy fine-tuning data, as well as several other instruction-tuning datasets, including ShareGPT-4V Chen et al. (2023b), LLaVA-OneVision Li et al. (2024a), ALLaVA-4V Chen et al. (2024a), and Cambrian-1 Tong et al. (2024). For a fair comparison, we randomly sampled the same number of samples from each dataset as in TaskGalaxy for fine-tuning. The results, summarized in Table A-5, demonstrate that TaskGalaxy consistently achieves the highest performance on most benchmarks across multiple model architectures, validating its effectiveness.

Table A-5: Experimental Evaluation of TaskGalaxy, Baseline Dataset, and Other Instruction-Tuning Datasets Separately. All the numbers are presented in % except MME and the full score is 100%. The indicator of MME is the perception score, the maximum value is 2000. The best results are highlighted in **bold**.

| Model | Method | Benchmarks | | | | | | | | |
|---|---|---|---|---|---|---|---|---|---|---|
| | | MME | MMB | MMB$^{CN}$ | POPE | LLaVA$^W$ | MMVet | TQA | SQA | MathVista |
| LLaVA-v1.5-7B | Baseline | 1476 | 63.29 | 56.45 | 86.30 | 47.70 | 24.70 | 57.59 | 68.77 | 28.20 |
| | ShareGPT-4V | 1501 | 65.97 | 59.10 | 86.29 | 49.20 | 29.00 | 57.56 | 70.60 | 28.20 |
| | LLaVA-OneVision | 1251 | 59.79 | 52.84 | 83.90 | 51.20 | **29.60** | 52.99 | **73.19** | 28.20 |
| | ALLaVA-4V | 1474 | 60.13 | 55.39 | 84.21 | 38.00 | 27.00 | 53.77 | 70.05 | 29.20 |
| | Cambrian-1 | 1494 | 61.08 | 54.46 | 85.46 | 52.00 | 25.70 | 55.17 | 71.03 | 29.10 |
| | TaskGalaxy | **1520** | **66.62** | **59.43** | **86.40** | 52.30 | 28.60 | **58.08** | 71.06 | **29.30** |
| | | ChartQA | AI2D | Q-Bench | Q-Bench$^{CN}$ | HalluBench | SEED | MMMU | Average (w/o MME) | |
| | Baseline | 14.40 | 25.29 | 24.89 | 31.26 | 47.95 | 58.62 | 19.70 | 43.62 | |
| | ShareGPT-4V | 17.84 | 27.08 | 26.22 | 32.51 | 48.79 | 59.26 | 15.60 | 44.48 | |
| | LLaVA-OneVision | 18.72 | 27.95 | 24.48 | 33.51 | 47.74 | 30.06 | 17.00 | 41.68 | |
| | ALLaVA-4V | 17.00 | 23.73 | 23.95 | 33.18 | 48.73 | 40.52 | 18.10 | 41.53 | |
| | Cambrian-1 | 20.07 | 29.46 | 26.70 | 33.61 | 50.78 | 49.52 | 19.30 | 44.23 | |
| | TaskGalaxy | 19.90 | **32.70** | **30.24** | **34.01** | 50.95 | 59.32 | **20.70** | **46.49** | |
| | | MME | MMB | MMB$^{CN}$ | POPE | LLaVA$^W$ | MMVet | TQA | SQA | MathVista |
| InternVL-Chat-v1.0-7B | Baseline | 1488 | 64.86 | 56.41 | 86.03 | 48.30 | 25.70 | 55.29 | 65.63 | 27.00 |
| | ShareGPT-4V | 1191 | 47.08 | 40.63 | 82.62 | 30.00 | 17.90 | 44.21 | 64.55 | 27.20 |
| | LLaVA-OneVision | 1350 | 61.23 | 54.74 | 67.94 | 32.50 | 19.40 | 37.74 | 66.29 | 25.00 |
| | ALLaVA-4V | 1425 | 62.76 | 52.78 | 84.50 | 21.50 | 23.50 | 48.04 | 66.29 | 29.40 |
| | Cambrian-1 | 1481 | 60.22 | 53.01 | 84.17 | 43.70 | 26.80 | 52.61 | 67.71 | **33.00** |
| | TaskGalaxy | **1512** | **65.03** | **57.91** | **86.23** | 52.30 | **30.10** | 56.15 | **68.88** | 30.10 |
| | | ChartQA | AI2D | Q-Bench | Q-Bench$^{CN}$ | HalluBench | SEED | MMMU | Average (w/o MME) | |
| | Baseline | 14.12 | 35.92 | 42.89 | 43.73 | 51.94 | 59.06 | 26.90 | 47.17 | |
| | ShareGPT-4V | 14.52 | 35.59 | 46.69 | 36.38 | 52.36 | 47.24 | 30.30 | 42.48 | |
| | LLaVA-OneVision | 13.76 | 22.75 | 40.08 | 42.89 | 53.39 | 40.87 | 24.60 | 40.20 | |
| | ALLaVA-4V | 12.99 | 28.28 | 42.87 | 44.16 | 51.41 | 48.36 | 27.30 | 42.94 | |
| | Cambrian-1 | **16.00** | 36.69 | 48.00 | 41.33 | **54.63** | 56.24 | 30.60 | 46.98 | |
| | TaskGalaxy | 15.16 | **37.69** | **48.21** | **46.32** | 53.00 | **60.44** | **32.80** | **49.63** | |

## A.8 More advanced model architecture

For more advanced models, we utilize the InternVL-Chat-V2.0-8B model, which has made its second-stage instruction fine-tuning data publicly available. For the comparison, we randomly sample the same number of samples as TaskGalaxy from the officially disclosed instruction fine-tuning dataset.

We fine-tune InternVL-Chat-V2.0-8B using both the original instruction fine-tuning dataset and the TaskGalaxy instruction fine-tuning dataset, ensuring that the number of samples for each is consistent with TaskGalaxy. The Table A-6 shows the performance comparison between the original instruction fine-tuning dataset and the TaskGalaxy instruction fine-tuning dataset on InternVL-Chat-V2.0-8B.

Table A-6: Experimental Evaluation of Fine-tuning InternVL-Chat-v1.0-8B Using TaskGalaxy and Baseline Dataset Individually. All the numbers are presented in % except MME and the full score is 100%. The indicator of MME is the perception score, the maximum value is 2000. The best results are highlighted in **bold**.

| Model | Method | Benchmarks | | | | | | | | |
|---|---|---|---|---|---|---|---|---|---|---|
| | | MME | MMB | MMB$^{CN}$ | POPE | LLaVA$^W$ | MMVet | TQA | SQA | MathVista |
| InternVL-Chat-V2.0-8B | Baseline | 1536 | 68.52 | 66.46 | 86.30 | **63.20** | 46.17 | 66.24 | 90.58 | 50.10 |
| | TaskGalaxy | **1565** | **73.88** | **70.79** | **86.90** | 62.85 | **48.86** | **70.49** | **92.71** | **52.31** |
| | | ChartQA | AI2D | Q-Bench | Q-Bench$^{CN}$ | HalluBench | SEED | MMMU | Average (w/o MME) | |
| | Baseline | **76.64** | 75.88 | 57.79 | 56.98 | 57.51 | 62.72 | 40.50 | 65.86 | |
| | TaskGalaxy | 76.56 | **76.75** | **59.65** | **57.12** | **58.99** | **64.25** | **41.22** | **67.81** | |

As shown in the Table A-6, after fine-tuning the model using the TaskGalaxy dataset and the original InternVL-Chat-V2.0-8B fine-tuned dataset, TaskGalaxy outperforms the original baseline dataset on 14 out of 16 benchmarks. For the remaining two benchmarks, ChartQA and LLaVA-in-the-wild, the performance difference compared to the baseline dataset is minimal (less than 0.5). This demonstrates that TaskGalaxy's enhancement of task diversity is also effective for more advanced models, providing a significant boost in overall performance.

## A.9 The benefits of Chain-of-Thought(CoT)

Numerous studies (Zhang et al., 2023b; Zheng et al., 2023; Zhang et al., 2024a; Mitra et al., 2024) have highlighted the significant impact of Chain-of-Thought (CoT) prompting on enhancing MLLM performance. In this section, we investigate whether CoT prompting improves performance with increased task types. We constrained the maximum number of samples per task type to 5, resulting in a total of 76k samples (as indicated by max_5 in Table A-7). We compared the performance using original TaskGalaxy Q&A data with CoT-generated answers from GPT-4o, designed through specific prompts detailed in the Appendix. The results show significant improvements for the CoT versions of TaskGalaxy (max_5) in benchmarks such as MME, LLaVA-in-the-wild, and Q-Bench. Additionally, the average performance across 15 benchmarks, excluding MME, increased by approximately 1.3 points with CoT. These findings underscore the value of incorporating CoT prompting into multimodal models.

Table A-7: **Performance comparison of CoT validity verification.** +max_5 refers to 19,227 task types, each containing no more than 5 samples, totaling approximately 76k samples. In contrast, +max_5 (CoT) represents the version where Chain-of-Thought (CoT) answers were generated for all the aforementioned samples.

| Model | Method | Benchmarks | | | | | | | |
|---|---|---|---|---|---|---|---|---|---|
| | | MME | MMB | LLaVA$^W$ | MathVista | ChartQA | Q-Bench | MMMU | Average( w/o MME) |
| LLaVA-v1.5-7B | Baseline | 1506 | 64.69 | 53.0 | 26.7 | 14.72 | 26.08 | 16.6 | 44.46 |
| | Baseline+max_5 | 1506 | 65.80 | 53.4 | 27.3 | 20.20 | 36.48 | 17.4 | 46.61 |
| | Baseline+max_5 (CoT) | 1523 | 66.72 | 64.7 | 27.9 | 20.96 | 43.27 | 19.3 | 47.92 |

## A.10 Task types in TaskGalaxy

One of the key challenges addressed by the TaskGalaxy instruction fine-tuning dataset is the substantial expansion of task type diversity. Initially, we manually defined a small set of task types,

which was later expanded to 19,227 hierarchical task types using GPT-4o. Given the large number of task types and space constraints, we present only a selection of these hierarchical task types here. A more comprehensive list will be available in the full dataset upon its release.

Table A-8: Comprehensive Task Type Table

| Level-1 Task Type | Level-2, etc. Task Types |
|---|---|
| OCR | bill ocr ~ medical bill recognition; bill ocr ~ travel expense recognition; book ocr ~ metadata extraction; book ocr ~ text summarization; business card ocr ~ text extraction; business card ocr ~ entity recognition; captcha ocr ~ multi-font text recognition; chart ~ bar chart ocr; chart ~ violin plot ocr; comic strip ocr ~ speech bubble detection; comic strip ocr ~ text translation; diagram ocr ~ flowchart recognition; document ~ form ocr; document ~ full text ~ literature ~ research paper ocr; font recognition ~ font type identification; graffiti ocr ~ text enhancement; grocery list ocr ~ item price extraction; handwritten text ocr ~ word recognition; infographic ocr ~ caption analysis; invoice ocr ~ data validation; invoice ocr ~ signature detection; label ocr ~ word recognition; lecture notes ocr ~ speaker identification and attribution; logo recognition ~ brand identification; logo recognition ~ logo location detection; musical notes ocr ~ time signature detection; ... |
| Image Description | abstract-concept-based ~ symbolic representation; abstract-concept-based ~ aesthetic judgment; accessibility-based description ~ highlight key elements; accessibility-based description ~ emotionally aware description; action-based ~ contextual action description; activity-based ~ sports event description; activity-based ~ performance or event description; advertisement-based ~ testimonial integration; art-style-based ~ surrealist description; attribute-based ~ pattern recognition; autobiographical-based ~ daily activities; bias-mitigation-based ~ disability bias detection; coarse-grained ~ highlight extraction; coarse-grained ~ alt text generation; context-aware description correction ~ factual accuracy verification; contrast-based ~ emphasis on unique aspects; cross-cultural adaptability ~ identify cultural context; culture-based ~ architecture and landmarks; culture-based ~ social norms and values; educational-content-based ~ step-by-step tutorial; emotion-based ~ emotion-driven storytelling; event-based ~ event sentiment analysis; ... |
| *Continued on next page* | |

| *Continued from previous page* | |
|---|---|
| Level-1 Task Type | Level-2, etc. Task Types |
| Detection | object detection ∼ single object detection ∼ firefighting equipment detection; object detection ∼ single object detection ∼ vehicle ∼ interior detection; target detection ∼ scene detection ∼ campus scene detection; target detection ∼ scene detection ∼ traffic scene detection; signature detection; out of stock detection; anomaly detection ∼ vehicle anomaly detection; anomaly detection ∼ behavior anomaly detection; object detection ∼ multiple object detection ∼ quantity detection; object detection ∼ single object detection ∼ road traffic signal detection; object detection ∼ single object detection ∼ public facility detection; target detection ∼ scene detection ∼ pond scene detection; signature detection; ... |
| Analysis | color analysis ∼ color contrast analysis; color analysis ∼ dominant color detection; complex scenes ∼ attribute extraction; content accessibility analysis ∼ automatic summarization for accessibility; content personalization ∼ user interest profiling; content prudence analysis ∼ source reliability evaluation; design concepts and intentions ∼ evaluate layout and composition; emotional analysis ∼ mood classification; emotional analysis ∼ emotion trajectory analysis; fashion analysis ∼ season identification; fashion analysis ∼ color analysis; game rules and strategies ∼ goal determination; gender representation analysis ∼ gender stereotype identification; gender representation analysis ∼ gender representation in advertising analysis; language use analysis ∼ language complexity analysis; language use analysis ∼ speech act recognition; review analysis ∼ review summarization; review analysis ∼ authenticity verification; political sentiment analysis ∼ polarization detection; political sentiment analysis ∼ emotion detection; ... |
| Image-based Knowledge Distillation | fine-grained image classification; semantic segmentation; caption generation; multimodal trend analysis; image inpainting; region-based image captioning; object attribute extraction; attribute recognition; content summarization; visual relationship detection;...... |
| *Continued on next page* | |

| Continued from previous page | |
|---|---|
| Level-1 Task Type | Level-2, etc. Task Types |
| Content Creation | FAQ creation ∼ Extract information from images to create detailed FAQ answers; FAQ creation ∼ Determine ambiguity in text and images for FAQ refinement; advertising content creation ∼ brochure content creation; advertising content creation ∼ flyer content creation; annual report writing ∼ industry benchmarking; annual report writing ∼ risk assessment; art critique ∼ technique and brushwork examination; art critique ∼ gesture and movement evaluation; artistic inspiration writing ∼ mood setting; artistic inspiration writing ∼ genre-specific style implementation; augmented reality content creation ∼ virtual staging and design; augmented reality content creation ∼ 3D object placement; brainstorming ∼ graphic design concepts; brainstorming ∼ product ideation; brainstorming ∼ headline generation; children's book creation ∼ conflict resolution; children's book creation ∼ narrative voice consistency; content curation ∼ content diversification; corporate training content creation ∼ employee onboarding material creation; diversity and inclusion content creation ∼ celebration of cultural events; e-commerce content creation ∼ purchase decision support content creation; e-commerce content creation ∼ content creation for limited-time promotions; educational game content creation ∼ image captioning games; interactive content creation ∼ immersive simulations; letter writing ∼ closing statement formulation; log writing ∼ daily summary; log writing ∼ task management; ... |
| Suggestions | home decor ideas; furniture design; wedding planning; recipe suggestions; seasonal decorations; party themes; workspace ergonomics; hair styling; book recommendations; interior lighting ideas; eco-friendly products; movie suggestions; fashion advice; life hacks; meditation practices; time management techniques; TV show recommendations; coding resources; online learning platforms; memory improvement exercises; volunteer opportunities; language translation aids; entertainment activities; speech writing; content creation tools; study techniques;...... |
| Continued on next page | |

| *Continued from previous page* | |
|---|---|
| Level-1 Task Type | Level-2, etc. Task Types |
| Subject Question and Answer | mathematics; transportation&logistics; religion; language& literature; law; agriculture; architecture; psychology; geography; environmental studies; political science; entertainment; economics; philosophy; education; history; media&communication; arts&arts~music studies; arts&arts~design studies; arts&arts~academic theoretical studies; business~management studies; business~finance studies; business~accounting studies; health&medicine~clinical medical studies; humanities&social sciences~history studies; science~geography studies; science~mathematics studies; technology&engineering~mechanical engineering studies; technology&engineering~electrical& energy studies;...... |
| Summarization | sports event summarization; pop culture summarization; policy summarization; news article summarization; book overview summarization; technical document summarization; agricultural data summarization; recipe summarization; presentation summarization; environmental impact summarization; genealogy summarization; customer journey summarization; financial report summarization; advertisement summarization; legal document summarization; travel itinerary summarization; instructional summarization; customer preferences summarization; market analysis summarization; political debate summarization; comparative summarization; multimodal dataset summarization; multilingual content summarization; event summarization; healthcare report summarization; wildlife monitoring summarization; customer service conversation summarization; event outcome summarization; climate data summarization; historical document summarization; educational content summarization; radio interview summarization; product description summarization; brand sentiment summarization; thematic summarization; visual trend summarization; image-based content summarization; product review summarization;...... |
| *Continued on next page* | |

| *Continued from previous page* | |
|---|---|
| Level-1 Task Type | Level-2, etc. Task Types |
| Logical Reasoning | abductive reasoning~multimodal hypothesis generation; analogy reasoning~scene relationship analogy reasoning; causal reasoning~counterfactual reasoning; complex reasoning~three-dimensional spatial relationship reasoning; complex reasoning~passenger flow analysis reasoning; complex reasoning~market analysis reasoning; complex reasoning~task relationship reasoning; complex reasoning~ethical dilemma reasoning; complex reasoning~role reasoning; complex reasoning~state reasoning; complex reasoning~chart understanding and analysis~multi-layer pie chart logical understanding and analysis; complex reasoning~chart understanding and analysis~high-dimension data scatter plot understanding and analysis; complex reasoning~complex mathematical calculations~geometric mathematical operations; complex reasoning~complex mathematical calculations~computer science algorithms operations; deductive reasoning~attribute deduction; deductive reasoning~inference of missing information; simple reasoning~goal-directed reasoning; simple reasoning~emotion reasoning; simple reasoning~sequence ordering; simple reasoning~basic mathematics~simple numerical calculation reasoning; simple reasoning~basic mathematics~quantity reasoning; spatial reasoning~topological reasoning; spatial reasoning~3D scene reconstruction; spatial reasoning~spatial layout recognition; temporal reasoning~contextual time inference; temporal reasoning~event ordering; ...... |
| Context-Aware Recommendations | seasonal and holiday recommendations; personalized shopping guides; context-based travel suggestions; context-specific personal assistant; visual product recommendations; real-time event notifications; location-based activity suggestions; dynamic content customization; context-sensitive educational content; contextual news delivery; adaptive learning resources; contextual target audience analysis; lifestyle-based content curation; interest-based content filtering; mood-based content suggestions; context-specific social media posts; personalized content suggestions; |
| *Continued on next page* | |

| Continued from previous page | |
| --- | --- |
| Level-1 Task Type | Level-2, etc. Task Types |
| Refusal | due to animal cruelty~identification of animal types in images; due to animal cruelty~image and text alignment for cruelty evidence; due to child exploitation~identification of minors; due to child exploitation~engagement with law enforcement; due to conspiracy theories~bi-modal sentiment analysis related to conspiracies; due to conspiracy theories~detect repetitive conspiracy motifs; due to dangerous stunts~compare depicted stunts with known dangerous activities; due to dangerous stunts~analyze risk levels of described actions; due to deepfake content~classification; due to human trafficking~detecting document forgery; due to incitement of panic~cross-verify with trusted sources; due to political propaganda~identify exaggeration in political claims; due to solicitation~detecting refusal language; refusal due to illegal activities~refusal due to illegal distribution; refusal due to pornographic content~characterization of suggestive poses;...... |
| 3D Object Recognition | Depth estimation~Self-supervised depth estimation; Material and texture recognition~Material texture segmentation in images; Material and texture recognition~Material texture correlation between images and text; Object attribute extraction~Color detection; Object detection~Real-time object detection; Object detection~Category-based object detection; Object interaction modeling~Object spatial relationships; Object matching and retrieval~Aligning textual descriptions with visual object attributes; Object matching and retrieval~Multi-view image retrieval from textual input; Object part recognition~Part-based object localization; Object part segmentation~Part relationship analysis; Object pose estimation~Instance-level pose estimation; Object recognition in context~Semantic segmentation of objects in context; Object registration~Multi-view registration; Object tracking~Object localization; Occlusion handling~Multi-view fusion for occluded objects; Occlusion handling~Self-supervised learning for occlusion robustness;...... |
| Continued on next page | |

| Continued from previous page | |
|---|---|
| Level-1 Task Type | Level-2, etc. Task Types |
| Safety and Risk Assessment | aviation safety and risk assessment~runway safety monitoring; aviation safety and risk assessment~air traffic control communication analysis; biological hazard risk assessment~foodborne pathogen identification; biological hazard risk assessment~water quality assessment; biological hazard risk assessment~disease outbreak identification; chemical hazard risk assessment~incident analysis and reporting; child safety and risk assessment~age-appropriate content detection; child safety and risk assessment~explicit content filtering; construction site safety and risk assessment~safety signage compliance; construction site safety and risk assessment~emergency response preparedness; consumer product safety and risk assessment~label product hazards; cultural heritage safety and risk assessment~damage assessment; cultural heritage safety and risk assessment~environmental hazard identification; elderly care safety and risk assessment~fall risk detection; entertainment venue safety and risk assessment~first aid station location identification; environmental safety and risk assessment~wildlife impact assessment; infrastructure safety and risk assessment~bridge stability analysis; wildlife safety and risk assessment~monitoring of wildlife health;...... |
| Image-Text Matching | image sequence~visual storytelling; image sequence~next image prediction; image sequence~story board generation; multiple images~event chronology; multiple images~relationship extraction; multiple images~collage interpretation; single image~visual reasoning; multiple images~visual consistency; multiple images~relationship extraction; question answer selection~visual entailment; question answer selection~scene-text based question answering; question answer selection~image-caption-based question answering;...... |
| Continued on next page | |

| Continued from previous page | |
|---|---|
| Level-1 Task Type | Level-2, etc. Task Types |
| Science-Related | anomaly detection in scientific data~detecting unexpected changes in medical imaging and diagnostic reports; chart and diagram interpretation~identify underlying assumptions in data; chart and diagram interpretation~compare different data sets; chart and diagram interpretation~explain scientific concepts; citations and influence analysis~citation context analysis; citations and influence analysis~historical citation trends analysis; common knowledge question and answer~correlate text to scientific imagery; common knowledge question and answer~classify scientific categories in images; common knowledge question and answer~compare scientific phenomena in different images; common knowledge question and answer~deduce outcomes based on visual experiments; concept drifts detection~alteration in hypothesis testing; concept drifts detection~shift in research focus; concept drifts detection~updates in scientific vocabularies; conclusion extraction~pattern recognition; content paraphrasing~explanatory paraphrasing; data correlation analysis~metadata extraction; data correlation analysis~information synthesis; data correlation analysis~context-aware filtering; error detection and correction~table data correctness; experiment hypothesis generation~interactive multimodal hypothesis testing; scientific argumentation analysis~confounding factor detection; visual reasoning~flowchart analysis; visual reasoning~equation-visual correlation;...... |
| Concept Extraction | attribute extraction~texture recognition; attribute extraction~spatial geometry determination; attribute extraction~affordance recognition; attribute extraction~object functionality identification; caption generation~geographical context captioning; contextual similarity~contextual relationship extraction; contextual similarity~cross-modal context expansion; image-text localization~caption region association; keyphrase extraction~hierarchical keyphrase extraction; keyphrase extraction~frequency-based keyphrase extraction; relationship extraction~agent-action relationship identification; relationship extraction~causal relationship identification; relationship extraction~object-action relationship identification; summary generation~balanced summary;...... |
| Continued on next page | |

| Continued from previous page | |
|---|---|
| Level-1 Task Type | Level-2, etc. Task Types |
| Interactive Art Creation | 3D model generation from text~3D object reconstruction; 3D model generation from text~Image-based texture generation; adaptive theme-based art expansion using text~context-aware embellishment of art based on text; adaptive theme-based art expansion using text~coherence evaluation across multi-modal elements; adaptive theme-based art expansion using text~hierarchical theme structuring from text; art critique and suggestion~color analysis; art critique and suggestion~contextual relevance; art critique and suggestion~technical proficiency critique; artistic scene composition~color palette matching; collaborative art creation with text~crowdsourced art projects; conceptual visualization from narrative text~poetry-inspired illustrations; context-aware art adaptation~object removal;...... |
| Medical Imaging Analysis | document summarization; anomalous region detection~lesion detection; anomalous region detection~deformation detection; anomalous region detection~obstruction detection; biomarker identification~predictive biomarker discovery; biomarker identification~radiomic feature extraction; clinical trial matching~Imaging Biomarker Identification; clinical trial matching~Clinical Trial Summarization; clinical trial matching~Patient Profile Construction; data annotation~disease classification; data annotation~report generation; disease diagnosis~eye disease detection; disease diagnosis~tumor classification; disease diagnosis~autoimmune disease detection; disease diagnosis~diabetes-related imaging analysis; functional mapping~biomarker identification; image classification~organ segmentation; image classification~tissue type classification; image classification~image quality assessment; medical captioning~anatomical structure captioning; medical captioning~diagnostic summary captioning; medical captioning~procedure description captioning; medical captioning~lesion detection captioning; patient outcome prediction~complication risk assessment; patient outcome prediction~disease progression prediction; surgical assistance~training and simulation; surgical assistance~augmented reality visualization; treatment planning~tumor localization; treatment planning~risk assessment;...... |
| Continued on next page | |

| *Continued from previous page* | |
|---|---|
| Level-1 Task Type | Level-2, etc. Task Types |
| Multimodal Translation | text-based image description; image to motivational quote generation; image to narrative generation; image to multilingual article generation; dynamic image caption generation; fine-grained image understanding and text generation; image to paragraph generation; image-based dialogue generation; contextual image description generation;...... |
| Multiple Choice Questions | text-based; image-based; combined media-based; image-text based~image focus; image-text based~balanced focus; image-text based~text focus; image-text based~contextual inference; image-text based~temporal understanding;...... |
| Scene Understanding | activity recognition~sleep behavior analysis; activity recognition~animal behavior recognition; activity recognition~gesture recognition; activity recognition~group activity recognition; aption generation~style-specific captioning; context reasoning~cultural context detection; event detection~event duration estimation; expression recognition~expression detection in group images; expression recognition~pose estimation for expressions; gesture recognition~gesture pose estimation; human-object interaction detection~interaction dynamics modeling; human-object interaction detection~interaction localization; human-object interaction detection~interactive object detection scene understanding~human-object interaction detection~relationship extraction; relationship detection~human-human relationship; relationship detection~human-environment relationship; relationship detection~human-object relationship; scene classification~urban scene classification; scene classification~commercial scene classification; scene classification~recreational scene classification; scene description~object identification; scene description~scene composition evaluation; scene description~background element identification; visual reasoning~spatial reasoning; visual reasoning~contextual reasoning;...... |
| *Continued on next page* | |

| *Continued from previous page* | |
|---|---|
| Level-1 Task Type | Level-2, etc. Task Types |
| Target Recognition in Special Image Domains | target recognition in paintings; target recognition in sketches; target recognition in clip art; target recognition in doodles; target recognition in low-resolution images; target recognition in photographs; target recognition in cartoons; target recognition in infographics; target recognition in x-ray images; target recognition in ct images; target recognition in 3d rendered images; target recognition in low exposure images;...... |
| Topic Classification | business~leadership changes; business~mergers and acquisitions; business~financial reporting; business~competitive analysis; business~corporate social responsibility; education~student proficiency assessment; education~educational resource identification; education~education level identification; emotion detection~anger detection; entertainment~event classification; entertainment~comic book identification; health~exercise and fitness tracking; health~treatment recommendation; health~mental health assessment; lifestyle~health and wellness; lifestyle~hobbies and crafts; politics~election event detection; politics~political sentiment analysis; politics~political stance detection; science~geology; science~computer science; sports~match predictions; sports~transfer news; technology~software categorization; technology~industry trends; travel~travel safety information extraction; travel~travel tips extraction;...... |

A.11  SAMPLES IN TASKGALAXY

In the main paper we provide a small number of sample question-answer pairs corresponding to task types, and in this section we provide more examples.

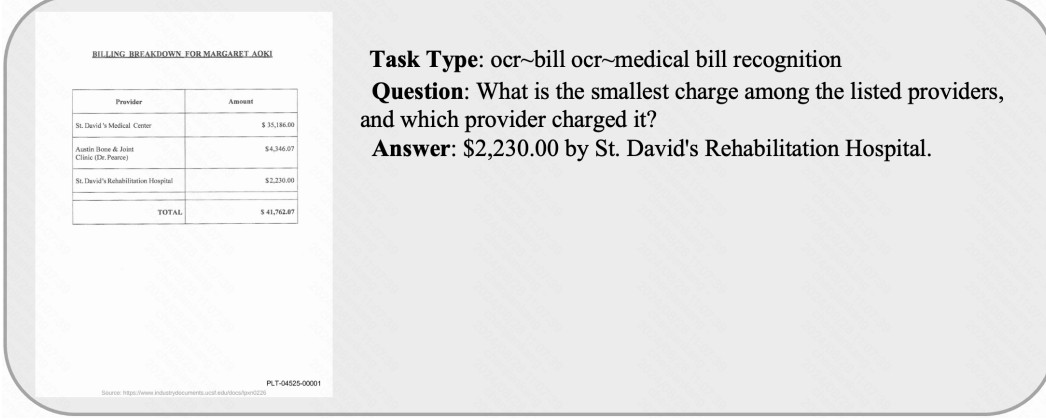

**Task Type**: ocr~bill ocr~medical bill recognition
**Question**: What is the smallest charge among the listed providers, and which provider charged it?
**Answer**: $2,230.00 by St. David's Rehabilitation Hospital.

Figure A-13: Task Type: ocr∼bill ocr∼medical bill recognition

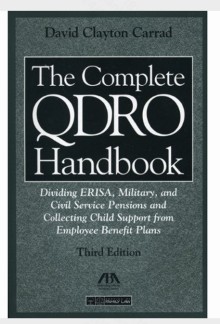

**Task Type**: ocr~book ocr~metadata extraction
**Question**:Who is the author of the book mentioned on the cover?
**Answer**: The author of the book is David Clayton Carrad.

Figure A-14: Task Type: ocr∼book ocr∼metadata extraction

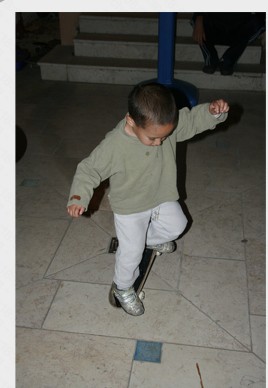

**Task Type**: image description~activity-based~sports event description
**Question**:Is the child participating in a sports activity, and how would you describe the context?
**Answer**: Yes, the child is engaged in a sports activity, specifically skateboarding. The context is likely casual practice indoors, suggesting the child is learning or playing at skateboarding rather than participating in a formal sports event.

Figure A-15: Task Type: image description∼activity-based∼sports event description

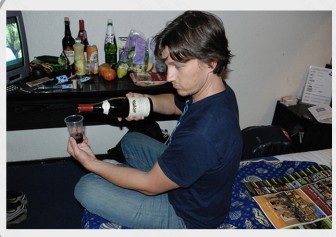

**Task Type**: image description~autobiographical-based~daily activities
**Question**:What daily activity is the person in the image engaged in?
**Answer**: The person in the image is engaged in the activity of pouring wine into a cup, which could be part of a social or personal relaxation activity.

Figure A-16: Task Type: image description~autobiographical-based~daily activities

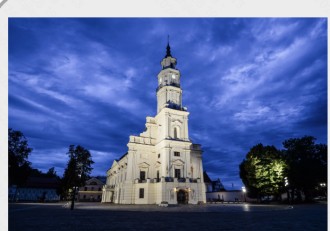

**Task Type**: image description~culture-based~architecture and landmarks
**Question**:Can you describe the architectural style and design elements of the building shown in the image?
**Answer**: The building in the image exhibits a blend of Gothic, Renaissance, and Baroque architectural styles. It features a tall, slender tower with decorative elements and windows, typical of Gothic and Renaissance architecture. The façade is adorned with classical Baroque details, including ornamental balconies and intricately designed cornices.

Figure A-17: Task Type: image description~culture-based~architecture and landmarks

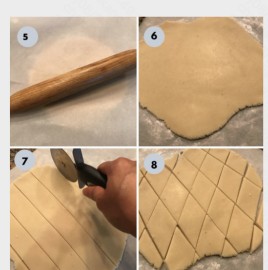

**Task Type**: image description~educational-content-based~step-by-step tutorial
**Question**:Provide a step-by-step tutorial based on the images for an educational content setting.
**Answer**: Step 1: Use a rolling pin to roll out the dough to an even thickness (image 6). Step 2: Take a pizza cutter and begin cutting the dough into long, even strips (image 7). Step 3: Rotate your cutting direction 90 degrees and cut again to form a grid pattern (image 8). Ensure the dough is evenly cut for uniform baking.

Figure A-18: Task Type: image description~educational-content-based~step-by-step tutorial

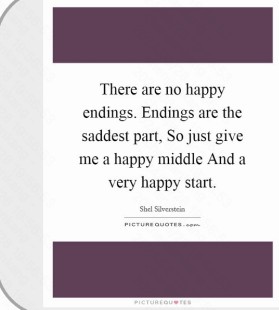

**Task Type**: image description~emotion-based~emotion-driven storytelling
**Question**:Describe the image as part of an emotion-driven story.
**Answer**: In the realm of literature, where Shel Silverstein's words weave emotions into the fabric of existence, the quote serves as a reminder. The people didn't fear the end, for they celebrated the beginnings with radiant smiles and lived the middles with unguarded joy, creating memories that made every end seem but a brief intermission in the ongoing narrative of happiness.

Figure A-19: Task Type: image description~emotion-based~emotion-driven storytelling

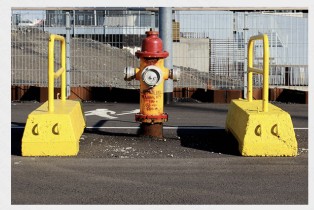

**Task Type**:detection~object detection~single object detection~firefighting equipment detection
**Question**:What firefighting equipment is shown in the image?
**Answer**:The image shows a fire hydrant.

Figure A-20: Task Type: detection~object detection~single object detection~firefighting equipment detection

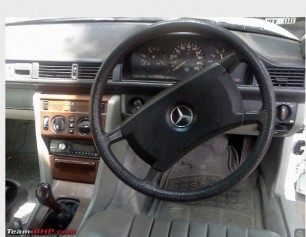

**Task Type**:detection~object detection~single object detection~vehicle~interior detection
**Question**:Which specific vehicle brand is detected in this image based on its interior components?
**Answer**:The vehicle brand detected based on its interior components, such as the steering wheel logo, is Mercedes-Benz.

Figure A-21: Task Type: detection~object detection~single object detection~vehicle~interior detection

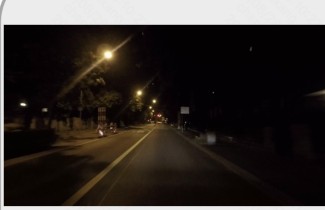

**Task Type**:detection~target detection~scene detection~traffic scene detection
**Question**:What kind of traffic scene is depicted in the image?
**Answer**:The image depicts a night-time urban street with some ongoing roadwork on the left side and a cyclist in the distance.

Figure A-22: Task Type: detection~target detection~scene detection~traffic scene detection

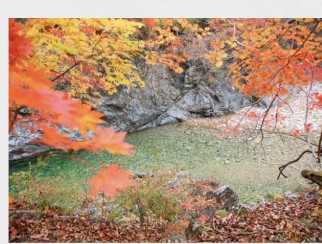

**Task Type**:analysis~color analysis~color contrast analysis
**Question**:What aspects of color contrast are significant in this image?
**Answer**:The significant aspect is the contrast between the warm, bright colors of the autumn foliage and the cool, neutral tones of the rocks and water. This contrast creates a visually appealing composition, drawing attention to the leaves while maintaining a natural, balanced setting.

Figure A-23: Task Type: analysis~color analysis~color contrast analysis

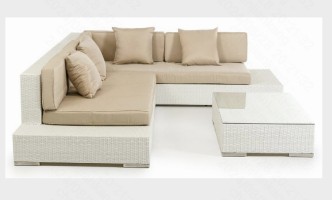

**Task Type**:analysis~design concepts and intentions~evaluate layout and composition
**Question**:What is the evaluation of the layout and composition of the furniture in the image?
**Answer**:The layout is efficient and cohesive, with the L-shaped sectional maximizing seating and the coffee table centrally placed for convenience.

Figure A-24: Task Type: analysis~design concepts and intentions~evaluate layout and composition

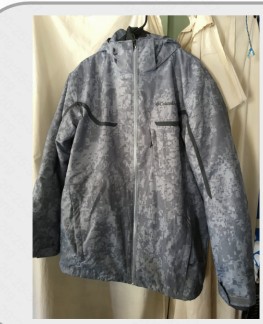

**Task Type**:analysis~fashion analysis~season identification
**Question**:For which season is this jacket most suitable?
**Answer**:This jacket is most suitable for winter or late fall, given its thick material and protective design.

Figure A-25: Task Type: analysis~fashion analysis~season identification

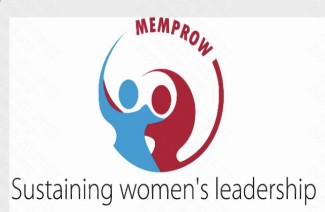

**Task Type**:analysis~gender representation analysis~gender representation in advertising analysis
**Question**:How does the image represent gender in the context of advertising?
**Answer**:The image represents gender in advertising by focusing on women's empowerment and the importance of female leadership.

Figure A-26: Task Type: analysis~gender representation analysis~gender representation in advertising analysis

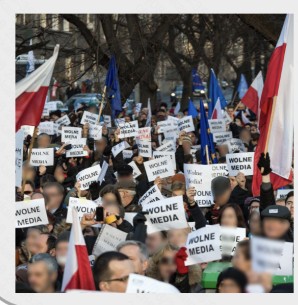

**Task Type**:analysis~political sentiment analysis~polarization detection
**Question**:analysis~political sentiment analysis~polarization detection
**Answer**:Yes, the image depicts a politically polarized environment, evidenced by a large group of protesters holding signs and flags, indicating a strong stance on media freedom which suggests a significant division in opinion.

Figure A-27: Task Type: analisis~political sentiment analysis~polarization detection

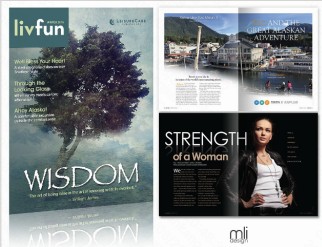

**Task Type**:content creation~advertising content creation~brochure content creation
**Question**:What type of imagery and content from the image would be effective in a promotional brochure?
**Answer**:Images of scenic travel destinations like Alaska and articles on wellness and personal empowerment would be engaging. These elements can visually appeal to prospects looking for enriching and adventurous experiences, reflecting Leisure Care's offerings.

Figure A-28: Task Type: content creation~advertising content creation~brochure content creation

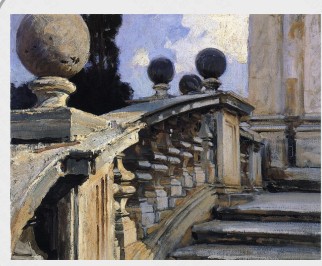

**Task Type**:content creation~art critique~technique and brushwork examination
**Question**:What can be inferred about the artist's brushwork and technique in this painting?
**Answer**:The artist employs a loose and expressive brushwork technique that captures the texture and form of the architectural elements. The thick and energetic strokes suggest a quick and confident application of paint, particularly noticeable in the depiction of the stone surface and the balustrades. This approach adds dynamism and a sense of movement to the otherwise static architectural subject.

Figure A-29: Task Type: content creation~art critique~technique and brushwork examination

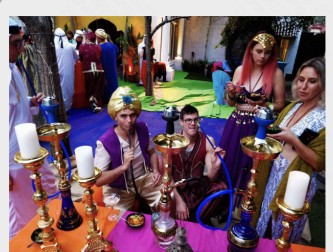

**Task Type**:content creation~diversity and inclusion content creation~celebration of cultural events
**Question**:How does this image exemplify diversity and inclusion in the celebration of cultural events?
**Answer**:The image exemplifies diversity and inclusion by showcasing a group of people from various backgrounds coming together to celebrate a cultural event, sharing in traditional practices, attire, and communal activities like smoking hookah and enjoying food.

Figure A-30: Task Type: content creation~diversity and inclusion content creation~celebration of cultural events

**Task Type**:content creation~letter writing~closing statement formulation
**Question**:What would be an effective closing statement to conclude the letter of recommendation for Denise Ball?
**Answer**:I strongly believe that Denise Ball would be a tremendous contribution to the Department of Education. She definitely has the knowledge, leadership qualities, and interpersonal skills that are needed to be successful.

Figure A-31: Task Type: content creation~letter writing~closing statement formulation

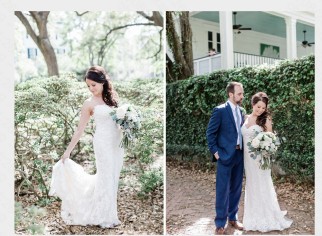

**Task Type**:suggestions~wedding planning
**Question**:How does the image provide inspiration for wedding planning?
**Answer**:The image inspires wedding planning by displaying a beautifully dressed bride and an elegantly dressed groom, set in scenic outdoor locations, suggesting a theme of an outdoor or garden wedding.

Figure A-32: Task Type: suggestions~wedding planning

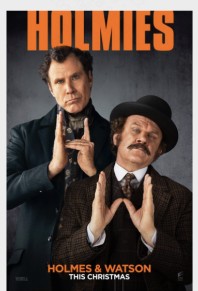

**Task Type**:suggestions~movie suggestions
**Question**:What movies would be good to watch if you enjoy 'Holmes & Watson'?
**Answer**:If you enjoy 'Holmes & Watson', you might also enjoy movies like 'Sherlock Holmes' (2009), 'Sherlock Holmes: A Game of Shadows' (2011), 'The Great Mouse Detective', and 'Enola Holmes'.

Figure A-33: Task Type: suggestions~movie suggestions

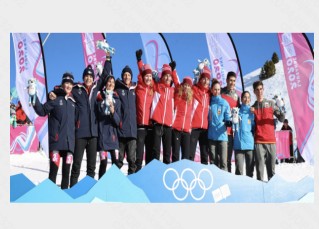

**Task Type**:summarization~sports event summarization
**Question**:Summarize the sports event displayed in the image.
**Answer**:The image shows a congratulatory moment where athletes from various countries celebrate their participation in a Winter Youth Olympic Games event, holding mascots and posing in front of Olympic symbols.

Figure A-34: Task Type: summarization~sports event summarization

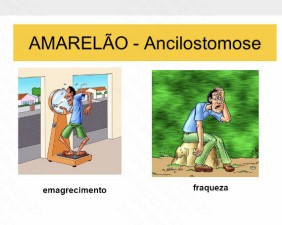

**Task Type**:logical reasoning~analogy reasoning~scene relationship analogy reasoning
**Question**:How are the scenes in the comic strip related by analogy?
**Answer**:Both scenes are visual analogies representing symptoms of ancylostomiasis. The first scene with the scale indicates extreme weight loss, while the second scene with the man sitting and holding his head depicts weakness and fatigue.

Figure A-35: Task Type: logical reasoning~analogy reasoning~scene relationship analogy reasoning

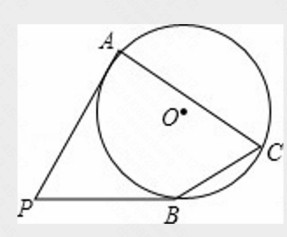

**Task Type**:logical reasoning~complex reasoning~complex mathematical calculations~geometric mathematical operations
**Question**:If O is the center of the circle and angle AOB is given as x degrees, what is the measure of angle ACB?
**Answer**:Angle ACB is half of angle AOB because an angle at the center of the circle is twice the angle at the circumference on the same arc. Therefore, ACB = x/2 degrees.

Figure A-36: Task Type: logical reasoning~complex reasoning~complex mathematical calculations~geometric mathematical operations

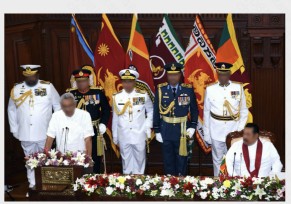

**Task Type**:logical reasoning~complex reasoning~military-related reasoning
**Question**:What can be inferred about the importance of the military in the context of this historical event?
**Answer**:The presence of multiple high-ranking military officials suggests that the military holds significant importance in the context of this historical event, possibly indicating a ceremony or announcement with major national security implications.

Figure A-37: Task Type: logical reasoning~complex reasoning~military-related reasoning

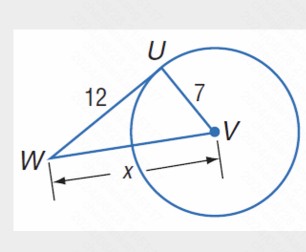

**Task Type**:logical reasoning~complex reasoning~complex mathematical calculations~geometric mathematical operations
**Question**:If the circle's radius is extended to 10, how will this affect the length of segment WV, given that the length of UW remains 12?
**Answer**:If the circle's radius VU is extended to 10 while UW remains 12, we use Pythagorean theorem again. UW^2 = WV^2 + VU^2. Substituting the new values: 12^2 = x + 10^2. So, 144 = x + 100. Therefore, x = 144 - 100 = 44. Thus, the length of WV when the circle's radius is extended to 10 is sqrt(44).

Figure A-38: Task Type: logical reasoning~complex reasoning~complex mathematical calculations~geometric mathematical operations

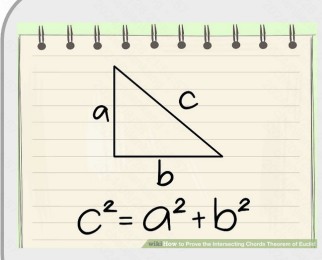

**Task Type**:logical reasoning~simple reasoning~basic mathematics~simple numerical calculation reasoning
**Question**:If a = 3 and b = 4, what is the value of c?
**Answer**:Using the Pythagorean theorem: c^2 = 3^2 + 4^2 = 9 + 16 = 25, hence c = 5.

Figure A-39: Task Type: logical reasoning~simple reasoning~basic mathematics~simple numerical calculation reasoning

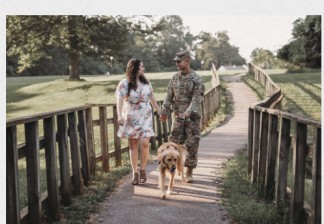

**Task Type**:logical reasoning~complex reasoning~military-related reasoning
**Question**:What can be inferred about the relationship between the two people based on the individual's attire and their interaction with each other?
**Answer**:The individual's military attire suggests that they are likely in a military service, and the casual attire and hand-holding suggest a personal, likely romantic relationship.

Figure A-40: Task Type: logical reasoning~complex reasoning~military-related reasoning

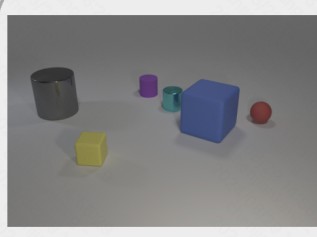

**Task Type**:logical reasoning~spatial reasoning~3D scene reconstruction
**Question**:How are the objects spatially arranged with respect to each other in the image?
**Answer**:The objects are spatially arranged as follows: The yellow cube is in the foreground to the left, the blue cube is the largest and is centrally placed to the right of the yellow cube, the red sphere is to the right of the blue cube, the gray cylinder is in the background to the far left, and two smaller cylinders (purple and turquoise) are between the gray cylinder and the blue cube, with the purple one being slightly farther back.

Figure A-41: Task Type: logical reasoning~spatial reasoning~3D scene reconstruction

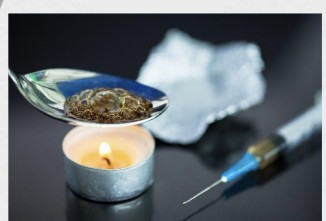

**Task Type**:refusal~refusal due to drug-related content~recognizing drug names and slang
**Question**:Can we recognize any drug names or slang in this image?
**Answer**:The image does not contain text or slang, but the visual elements suggest heroin or similar injectable drugs.

Figure A-42: Task Type: refusal~refusal due to drug-related content~recognizing drug names and slang

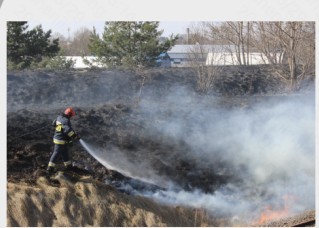

**Task Type**:safety and risk assessment~fire safety and risk assessment~post-incident damage assessment
**Question**:What evidence from the image can be used to assess the extent of damage caused by the wildfire?
**Answer**:The charred and blackened areas of the ground and vegetation indicate significant burning, suggesting a considerable extent of damage to the affected area.

Figure A-43: Task Type: safety and risk assessment~fire safety and risk assessment~post-incident damage assessment

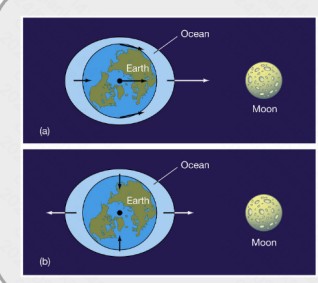

**Task Type**:science-related~image-text alignment~labeling diagrams with text explanations
**Question**:Label the key components and forces shown in both diagrams.
**Answer**:In both diagrams: The 'Earth' is labeled at the center. The 'Ocean' surrounds the Earth, represented by the light blue areas. The 'Moon' is positioned to the right, exerting a gravitational pull. Arrows indicate the direction of water movement: in (a), toward and away from the Moon, showing high tides; in (b), perpendicular to the Moon-Earth line, indicating low tides.

Figure A-44: Task Type: science-related~image-text alignment~labeling diagrams with text explanations

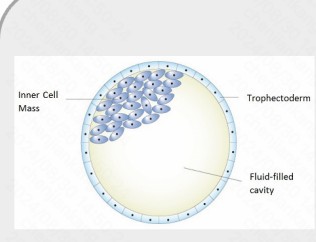

**Task Type**:science-related~image description generation~biological process illustration
**Question**:Describe the biological process that this image illustrates.
**Answer**:This image illustrates a blastocyst, an early stage of embryonic development. The blastocyst contains an outer layer of cells called the trophectoderm, an inner cell mass that will develop into the embryo, and a fluid-filled cavity known as the blastocoel.

Figure A-45: Task Type: science-related~image description generation~biological process illustration

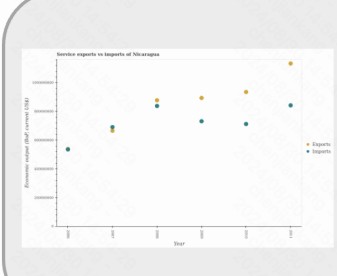

**Task Type**:science-related~data visualization interpretation~error bar interpretation
**Question**:What can be inferred about the relationship between service exports and imports of Nicaragua from 2006 to 2011?
**Answer**:From 2006 to 2011, both service exports and imports of Nicaragua show an upward trend. However, exports consistently surpass imports throughout the period.

Figure A-46: Task Type: science-related~data visualization interpretation~error bar interpretation

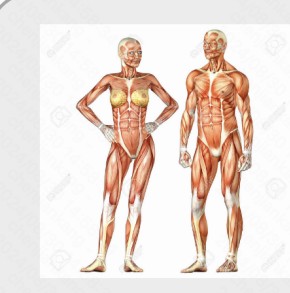

**Task Type**: science-related~visual reasoning~anatomical diagram labeling
**Question**: Label the major muscle groups shown on the figures.
**Answer**: In the male and female figures, the major muscle groups include the pectoralis major (chest), rectus abdominis (abs), external oblique (sides), quadriceps (front thigh), deltoid (shoulder), biceps brachii (arm), and gastrocnemius (calf muscles).

Figure A-47: Task Type: science-related~visual reasoning~anatomical diagram labeling

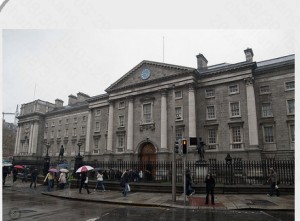

**Task Type**:subject question and answer~science~geography studies
**Question**:Explain how soil creep affects the landscape.
**Answer**:Soil creep affects the landscape by causing gradual downhill movement of soil and rock material. This process can lead to curved tree trunks, tilted fence posts, and distorted roadways as the ground slowly shifts over time.

Figure A-48: Task Type: subject question and answer~science~geography studies

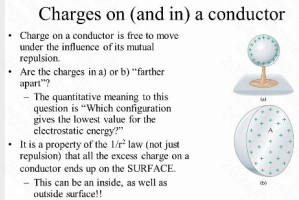

**Task Type**: subject question and answer~humanities & social sciences~history studies
**Question**: Can you name a famous historical figure associated with the institution shown in the image?
**Answer**: A famous historical figure associated with Trinity College Dublin is Jonathan Swift, the author of 'Gulliver's Travels', who was a student there in the late 17th century.

Figure A-49: Task Type: subject question and answer~humanities & social sciences~history studies

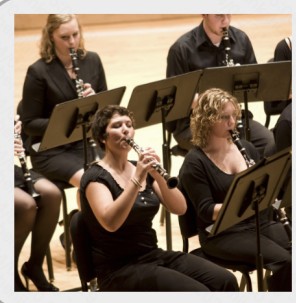

**Task Type**:subject question and answer~science~physics studies
**Question**:How does the spacing of charges differ between configurations (a) and (b) in the image?
**Answer**:In configuration (a), the charges are more concentrated because the spherical shape minimizes surface area for a given volume. In configuration (b), the charges are farther apart to cover a larger surface area, which helps to distribute the charges evenly and minimize electrostatic repulsion.

Figure A-50: Task Type: subject question and answer~science~physics studies

**Task Type**: subject question and answer~arts & arts~music studies
**Question**: In the context of an orchestra, what role does the clarinet section typically play?
**Answer**: In an orchestra, the clarinet section often provides rich and melodic mid-range tones, capable of versatile dynamics and expression, bridging the higher flutes and lower brass and strings.

Figure A-51: Task Type: subject question and answer~arts & arts~music studies

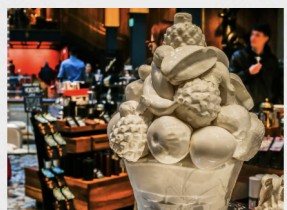

**Task Type**:3D object recognition~Material and texture recognition~Material wear and damage recognition from images
**Question**:What is the material of the sculpture in the image?
**Answer**:The sculpture is made of ceramic.

Figure A-52: Task Type: 3D object recognition~Material and texture recognition~Material wear and damage recognition from images

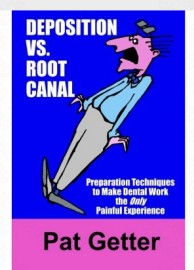

**Task Type**: concept extraction~text classification~humor detection
**Question**: What is the main theme of the image based on the text and visual cues?
**Answer**: The main theme of the image is comparing the discomfort of a legal deposition to a root canal, with a humorous tone focusing on minimizing the unpleasantness of dental work.

Figure A-53: Task Type: concept extraction~text classification~humor detection

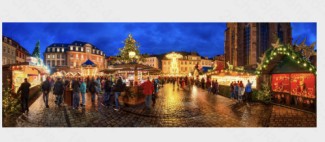

**Task Type**:context-aware recommendations~seasonal and holiday recommendations
**Question**:What recommendations would you provide for someone visiting this seasonal market for the first time?
**Answer**:For first-time visitors, it's recommended to explore the market during the evening to fully experience the festive lights. Start with a walk through the market to get a feel of the different stalls, try some local holiday treats, take plenty of photos of the lit decorations and historic buildings, and consider purchasing unique handcrafted gifts.

Figure A-54: Task Type: context-aware recommendations~seasonal and holiday recommendations

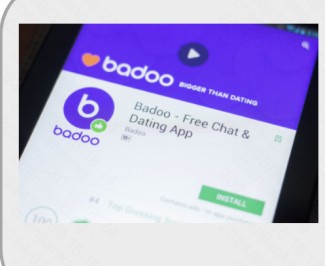

**Task Type**: context-aware recommendations~contextual chatbot responses
**Question**: What should a chatbot response be if a user asks about what makes Badoo different from other dating apps?
**Answer**: Badoo stands out by being more than just a dating app; it's a community where you can make real connections. It offers a unique blend of features that help you meet new friends, date, and even network. Plus, with our robust safety measures and a diverse user base, you're in safe hands.

Figure A-55: Task Type: context-aware recommendations~contextual chatbot responses

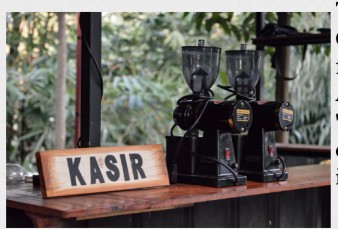

**Task Type**:image-based knowledge distillation~object recognition
**Question**:Identify the objects present in the image and their primary function.
**Answer**:The image shows two coffee grinders and a sign reading 'KASIR,' which typically indicates a cashier's area. The primary function of the coffee grinders is to grind coffee beans, and the 'KASIR' sign indicates where customers can pay for their purchases.

Figure A-56: Task Type: image-based knowledge distillation~object recognition

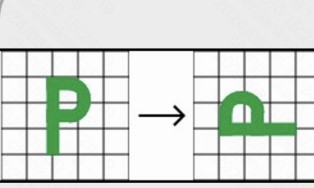

**Task Type**: image-text matching~single image~visual reasoning
**Question**: Based on the visual evidence, how would you describe the transformation that occurs in the image?
**Answer**: The letter 'P' on the left undergoes a 90-degree clockwise rotation to become a rotated form that no longer serves its original function as a letter but instead becomes an abstract shape.

Figure A-57: Task Type: image-text matching~single image visual reasoning

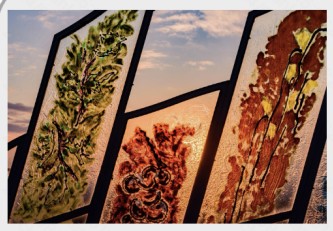

**Task Type**:interactive art creation~environment design based on textual lore~narrative coherence verification
**Question** : How can the stained glass art be used to create an environment inspired by a fantasy lore?
**Answer**:The stained glass can be used in the design of a grand hall or temple within a fantasy lore setting, where the shifting sunlight streaming through the colored glass creates a magical, otherworldly atmosphere.

Figure A-58: Task Type: interactive art creation~environment design based on textual lore~narrative coherence verification

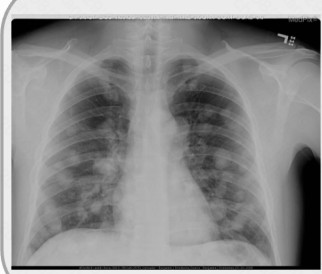

**Task Type**: medical imaging analysis~content extraction~disease identification
**Question**:Which specific lung condition can be identified from the chest X-ray image?
**Answer**:The image indicates the presence of pneumonia, characterized by the consolidations and opacities seen in the lungs.

Figure A-59: Task Type: medical imaging analysis~content extraction~disease identification

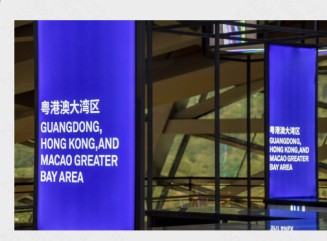

**Task Type**:multimodal translation~text-based image description
**Question** : Provide a detailed explanation of the signboard text and its visual features.
**Answer**: The image features two illuminated vertical signboards displaying bilingual text. The Chinese characters '粤港澳大湾区' are in large white font on a blue background. Below the Chinese text, the English translation 'GUANGDONG, HONG KONG, AND MACAO GREATER BAY AREA' is also displayed in white font. The signboards are positioned within a modern architectural environment.

Figure A-60: Task Type: multimodal translation~text-based image description

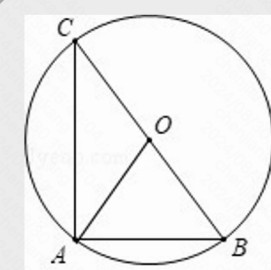

**Task Type**: multiple choice questions~image-text based~image focus
**Question**: If the diameter of the circle is 20 units, what is the length of AO?
A) 5 units B) 10 units C) 15 units D) 20 units
**Answer**: B) 10 units

Figure A-61: Task Type:multiple choice questions~image-text based~image focus

