# OpenReview forum: "TaskGalaxy: Scaling Multi-modal Instruction Fine-tuning with Tens of Thousands Vision Task Types"
_ICLR.cc/2025/Conference — ICLR 2025 Poster_

### Official Review · Reviewer_EFVj · 2024-10-31

**Soundness:** 3
**Presentation:** 3
**Contribution:** 3
**Rating:** 6
**Confidence:** 4

**Summary:**

This paper presents TaskGalaxy, a multi-modal instruction fine-tuning dataset. A pipeline for the systematic construction and generation of a diverse range of task types and corresponding high quality instruction Q&A samples is proposed. The fintuning results of LLaVA-V1.5 and InternVL-Chat-V1.0 validate the effectiveness of the proposed dataset.

**Strengths:**

1. The dataset construction pipeline is elegant and makes sense.
2. 19,227 hierarchical task types and 413,648 associated samples are a lot, I believe it can make contributions to the community.

**Weaknesses:**

1. As a submission on the area of datasets and benchmarks, the access to data is not provided to reviewers, which is obviously unreasonable.
2. The experiments are insufficient and lack comparison with other similar datasets.

**Questions:**

1. Compared with other existing similar datasets,  what are the advantages of the proposed dataset?
2. The comparisons of SFT results on various architectures with other similar datasets are suggested to supplement.
3. In the stage of match and filter, which CLIP model is adopted?
4. I wonder how the accuracy changes under the same architecture but with various sizes trained with the proposed dataset.

**Details Of Ethics Concerns:**

1. Supplement the License of image source in Table 1.
2. Which License will be adopted if the proposed dataset is open sourced.

---

> ### Author Response · Authors · 2024-11-22
>
> Dear Reviewer EFVj：
>
> Thank you for your constructive feedback and greatly appreciate your positive comments.
> We apologize for not providing reviewers with a link to TaskGalaxy. The link to the dataset is provided below: [TaskGalaxy Link](https://drive.google.com/drive/folders/1_RuE_U9ZyLMYvn7714-lKMOyMse2t76r?usp=drive_link)
> # 1.About advantages of the proposed dataset
>
> To explore the advantages proposed for our dataset, first of all, we would like to explain the motivation for generating this dataset: because the data requirements for fine-tuning the current large multimodal models are the quality, quantity, and diversity of the data, and we are more concerned about the diversity and quality of the data, which directly determines the generalization capability of the multimodal models, and the diversity of the data is reflected in the diversity of the tasks, and we start from the perspective of the task diversity to construct a high-quality multimodal instruction fine-tuning dataset that covers as many task types as possible in natural world scenarios.
>
> In order to explore the diversity of tasks, there are some previous works, but the generation of task types is very time-consuming and labor-intensive, and requires experts to manually generate and annotate, etc., and the generation of task types is limited to about 200 at most, which is difficult to cover the variety of task types in the open world. To address the above problems on the one hand, in order to liberate manpower as much as possible and generate as many task types as possible in the open world, the pipeline of our data generation is to utilize a few hierarchical task types generated by human beings as the initial seed, and then build a detailed prompt to let the robust closed-source model GPT-4o generate the task types hierarchically in a cyclic manner, which liberates manpower, and the task types are also limited by the prompt, and the task types are also limited by the prompt. The task types are also restricted to not overlap as much as possible by the prompt, so that the quantity and quality of the generated hierarchical task types are guaranteed.
>
> In addition, we designed a two-stage task type and open source image matching strategy, as well as a joint filtering strategy of multiple open source models on the question-answer pairs related to the generated task types to further ensure the quality of the data. Our generation pipeline takes into account both task diversity and data quality. Moreover, by utilizing our generation strategy, it is theoretically possible to flexibly expand the task types and question-answer samples. Our data generation strategy and the generated dataset can provide a reference for the instruction fine-tuning dataset community.
>
>
>
> # 3. About the CLIP model
>
> In the stage of match and filter,  CLIP-L/14 published by openAI is adopted. This model uses a ViT-L/14 Trasnformer architecture as an image encoder and uses a masked self-attention Transformer as a text encoder. These encoders are trained to maximize the similarity of (image, text) pairs via a contrastive loss. So we utilize it for the initial match and filter of every image and task type pools.
> # 4. About the acc change on various architecture
>
> In Table 2 of the article, we compared the models using two different architectures for each model across two architecture sizes. After incorporating the fine-tuning of our proposed dataset and excluding MME and POPE, which are not applicable to these two rubrics, the average accuracy for the remaining benchmarks is as follows: for llava-v1.5-7b and llava-v1.5-13b, the scores are 46.26 and 46.39, respectively. For specific benchmarks, the results are: MMB's 68.04 and 69.85, LlaVAw's 56.3 and 63.1, MMVet's 29.9 and 34.4, HalluBench's 51.74 and 53.21, and SEED's 60.28 and 61.22. A similar trend is observed for internevl-chat-v1.0-7b and -13b, indicating that a larger number of parameters in model training corresponds to higher accuracy, which aligns with our intuition.
>
> About details of ethics concerns:
> 1. The following additions have been made to the licensing information for image sources in Table 1 and will be incorporated in subsequent releases:
> | Dataset        | License          |
> |----------------|-------------------|
> | ALLaVA         | Apache License 2.0|
> | VisualGenome   | CC BY 4.0         |
> | MathV360K      | Apache License 2.0|
> | ShareGPT4V     | CC BY-NC 4.0      |
>
> 2. The license designated for the subsequent open-sourcing of the proposed dataset is CC BY-NC 4.0.

---

> ### Author Response · Authors · 2024-11-22
>
> # 2. About comparisons of SFT results on various architectures with other similar datasets
>
> To further validate the effectiveness of TaskGalaxy and to supplement the performance comparison with similar datasets, we conducted second-stage fine-tuning experiments using four datasets: TaskGalaxy, LLaVA-665K, LLaVA-OneVision, and ShareGPT4v. These datasets were used to fine-tune two models: LLaVA-v1.5-7b and InternVL-chat-v1.0-7b.
> To ensure fairness, we randomly sampled an equal number of data samples from the latter three datasets to match the size of TaskGalaxy. This approach highlights the impact of TaskGalaxy's rich and diverse task types. The experimental results are summarized in the following table:
>
> LLaVA-v1.5-7b:
> | **Benchmarks**        | **LLaVA-665K** | **LLaVA-OneVision** | **ShareGPT-4V** | **TaskGalaxy** |
> |-----------------------|----------------|---------------------|-----------------|----------------|
> | **MME**              | 1476  | 1251| 1501   | **1520**       |
> | **MMBench**          | 63.29          | 59.79    | 65.97           | **66.62**      |
> | **MMBench_CN**       | 56.45   | 52.84 | 59.10           | **59.43**      |
> | **TQA**              | 57.59  | 52.99   | 57.56    | **58.08**      |
> | **SQA**              | 68.77   | **73.19**           | 70.60   | 71.06          |
> | **ChartQA**          | 14.40          | 18.72   | 17.84           | **19.90**      |
> | **AI2D**             | 25.29  | 27.95   | 27.08  | **32.70**      |
> | **Q-Bench**          | 24.89   | 24.48   | 26.22 | **30.24**      |
> | **Q-Bench_CN**       | 31.26  | 33.51  | 32.51   | **34.01**      |
> | **HalluBench**       | 47.95  | 47.74   | 48.79   | **50.95**      |
> | **SEED**             | 58.62   | 30.06   | 59.26  | **59.32**      |
> | **MathVista**        | 28.20  | 28.20   | 28.20   | **29.30**      |
> | **MMMU**             | 19.70 | 17.00    | 15.60  | **20.70**      |
> | **POPE**             | 86.30          | 83.90               | 86.29    | **86.40**      |
> | **LLaVA-in-the-wild**| 47.70          | 51.20  | 49.20           | **52.30**      |
> | **MMVeT**            | 24.70          | **29.60**           | 29.00           | 28.60          |
> | **Average (w/o MME)**| 43.62          | 41.68 | 44.48           | **46.49**      |
>
> InternVL-Chat-V1.0-7b:
>
> | **Benchmarks**        | **LLaVA-665K** | **LLaVA-OneVision** | **ShareGPT-4V** | **TaskGalaxy** |
> |-----------------------|----------------|---------------------|-----------------|----------------|
> | **MME**              | 1488           | 1350                | 1191            | **1512**       |
> | **MMBench**          | 64.86          | 61.23               | 47.08           | **65.03**      |
> | **MMBench_CN**       | 56.41          | 54.74               | 40.63           | **57.91**      |
> | **TQA**              | 55.29          | 37.74               | 44.21           | **56.15**      |
> | **SQA**              | 65.63          | 66.29               | 64.55           | **68.88**      |
> | **ChartQA**          | 14.12          | 13.76               | 14.52           | **15.16**      |
> | **AI2D**             | 35.92          | 22.75               | 35.59           | **37.69**      |
> | **Q-Bench**          | 42.89          | 40.08               | 46.69           | **48.21**      |
> | **Q-Bench_CN**       | 43.73          | 42.89               | 36.38           | **46.32**      |
> | **HalluBench**       | 51.94          | **53.39**           | 52.36           | 53.00          |
> | **SEED**             | 59.06          | 40.87               | 47.24           | **60.44**      |
> | **MathVista**        | 27.0           | 25.0                | 27.2            | **30.1**       |
> | **MMMU**             | 26.9           | 23.3                | 30.3            | **32.8**       |
> | **POPE**             | 86.03          | 67.94               | 82.62           | **86.23**      |
> | **LLaVA-in-the-wild**| 48.3           | 32.5                | 30.0            | **52.3**       |
> | **MMVeT**            | 25.7           | 19.4                | 17.9            | **30.1**       |
> | **Average (w/o MME)**| 47.17          | 40.10                | 42.48           | **49.63**      |
>
> We sincerely appreciate your valuable comments. We have conducted more experiments to thoroughly address your concerns, and your suggestions and questions have been highly valuable in enhancing our work. Welcome any additional discussions or feedback.

---

> > ### Comment · Reviewer_EFVj · 2024-11-27
> > **Official Comment by Reviewer EFVj**
> >
> > Thanks,  my concerns are well resolved.

---

> > > ### Author Response · Authors · 2024-11-27
> > >
> > > We sincerely appreciate your thoughtful feedback and the increased score for our work. We will incorporate your suggestions into the next version of our paper to further improve and refine our work. Thank you once again for your valuable time and insightful comments.

---

### Official Review · Reviewer_VWt8 · 2024-11-01

**Soundness:** 3
**Presentation:** 3
**Contribution:** 3
**Rating:** 8
**Confidence:** 3

**Summary:**

The paper introduces a finetuning dataset for Vision-Language finetuning called TaskGalaxy, consisting of 19k hierarchical task types. The data is synthetically generated starting from seed tasks and contains 413k samples.
- First two levels of hierarchy are defined manually, and GPT-4o is prompted to expand the hierarchy
- The task type names are used to match images from multiple open source datasets using CLIP to match the text to images
- Given the top 10 matched task types to a given image, GPT-4o is used to filter out the irrelevant task types
- Based on the image and task types, GPT-4o is prompted to generate question-answer pairs
- Finally, 3 open source models (GLM-4v-9B, InternVL-Chat-V1.5 and InternVL2-26B) are prompted to score the generated question, task type and image with a score of 1 if the question and task type match relative to the image, and 0 otherwise. Samples are filtered through majority voting.
- The generated dataset is used to finetune models on downstream tasks, showing improved performance.

**Strengths:**

1. The paper proposes a scalable synthetic data generation pipeline across multiple task types from publicly available datasets.
2. Models finetuned with the generated dataset in addition to the original datasets show improved performance, which shows the effectiveness of the proposed dataset.
3. The dataset is ablated well to check the effect of varying the number of task types and number of samples for each task type, with higher number of task types and samples showing better performance, ~5% improvement on average (Table 2 and Figure 6).
4. The paper proposes a multi-stage pipeline for filtering the genearated samples, which can reduce the noise on the generated data.

**Weaknesses:**

1. **Check for hallucinations and incorrect responses**
- The paper does not have an explicit check for hallucinations and incorrect responses. Prior LLM generated datasets have had problems with hallucinations, leading to subpar performance on downstream tasks. Upon resolving the problems, the performance on downstream tasks typically increases [1]. While the multi-stage pipeline can potentially eliminate hallucinations, especially when 3 models are used as validators, the method and dataset will be more trustworthy if an explicit check is performed.
2. **Scope and impact**
- In the current form, the proposed dataset and generation pipeline is limited to building a good finetuning/visual instruction tuning dataset.
- The paper does not provide more information about the samples that are rejected during each stage of the filtering pipeline. The rejected samples might provide more insights into the behavior of the generator model and could be potentially valuable for future research into synthetic data generation. Studying this aspect might help broaden the impact of the work.
- For example, the authors could analyze type of question-answer pairs are typically rejected by the three filtering models, or if there is a pattern to which task types are rejected the most.
- This might also partially address Point 1 about hallucinations.

---

## References
[1] Gene Ruebsame, "Cleaned Alpaca Dataset: https://github.com/gururise/AlpacaDataCleaned", 2023.

**Questions:**

Please refer to the Weaknesses.

---

> ### Author Response · Authors · 2024-11-22
>
> Dear Reviewer VWt8,
>
> We sincerely appreciate your positive feedback and constructive suggestions regarding our work. We understand your interest in the presentation and analysis of incorrect and illusory samples generated by the MLLM model during the dataset creation process. Additionally, you highlighted the importance of showcasing the samples that were discarded through our screening and matching strategy at various stages. We agree that including examples of samples screened out for containing illusory or incorrect responses would greatly enhance the reader's understanding of our dataset generation process.
>
> In the field of multimodal large model fine-tuning, both data quality and diversity are crucial factors at this stage. A high-quality, diverse dataset with rich command fine-tuning capabilities is essential, and our work is centered around this focus. As you mentioned, the rejected samples in the second point partially correspond to those in the first point due to the illusions and erroneous responses generated by the MLLM during text generation. Here, we illustrate and analyze examples of samples that were filtered out throughout our entire data generation pipeline. For a detailed example, please refer to the PDF available at the anonymous link provided below:
> ***
> [Sample Analysis PDF](https://drive.google.com/file/d/1ryWlHlqSuETProbqcdPrpmRuu4R0pES5/view?usp=drive_link)
> ***
>
> Thank you very much for your suggestion. Analyzing the samples filtered out in the data generation pipeline provides valuable insights into the quality of TaskGalaxy and helps to better understand the current challenges faced by MLLMs in data generation.

---

> > ### Comment · Reviewer_VWt8 · 2024-11-26
> >
> > Thank you for your comment. The analysis of the rejected samples is indeed interesting. I would recommend the authors add this to the main paper's appendix and refer to it in the main text. This will help future research into synthetic data generation pipelines.
> >
> > I will raise my score to 8.

---

> > > ### Author Response · Authors · 2024-11-27
> > >
> > > Thank you for your response and thoughtful suggestion.
> > >
> > > We sincerely appreciate your recommendation to include the analysis of rejected samples in the appendix and reference it in the main text. This is indeed a valuable addition, and we will make sure to incorporate it in the revised version to benefit future research into synthetic data generation pipelines.
> > >
> > > Thank you once again for your constructive feedback and for raising your score. We are truly grateful for your time and insights.

---

### Official Review · Reviewer_EeP4 · 2024-11-02

**Soundness:** 2
**Presentation:** 3
**Contribution:** 2
**Rating:** 5
**Confidence:** 4

**Summary:**

This manuscript introduces a novel multi-modal instruction fine-tuning dataset called TaskGalaxy. TaskGalaxy proposes an automated pipeline comprising five main steps, which leverages robust multimodal model and reduces manual intervention. The proposed TaskGalaxy consists of tens of thousands vision task types and 413k samples to address the limitation of task diversity in existing datasets. The performance of LLaVA-V1.5 and InternVL-Chat-V1.0 model incorporating TaskGalaxy fine-tuning is improved across 16 benchmarks.

**Strengths:**

1.	The paper is easy to follow. The details of data construction and experiments are clear.

2.	TaskGalaxy contains a rich set of task types, alleviating the impact of limited task types on model generalization ability.

3.	The pipeline of TaskGalaxy can be flexibly expanded to create more high-quality data.

4.	Incorporating this training data, the performances of LLaVA-V1.5-13b and InternVL-Chat-V1.0-7b have been improved.

**Weaknesses:**

1.	In Hierarchical Task Type Generation, this process resulted in the generation of 19,227 hierarchical task types. But the task types are completely GPT-4o generated, how to ensure that 19,227 task types do not overlap with each other? The diversity of the tasks and data are neither thoroughly nor rigorously studied. There is no in-depth comparison to previous tasks and data.

2.	Since the improvements are obtained by training on baseline data + TaskGalaxy data, it is not possible to tell whether the improvements are due to the larger data amount or the larger task number.

3.	In Match and Filter section, images and task types are encoded by the off-the-shelf CLIP respectively to get the embeddings. As CLIP is trained mostly on data of natural concepts, filtering images with abstract phrases such as task type may be not as effective as expected.

4.	The performance of the baseline is too weak to prove the validity of the dataset. For example, InternVL-Chat-V1.0-7b only achieved the performance of 35.96% on AI2D and 15.2% on ChartQA. More advanced models, such as LLaVA-V1.6 and InternVL-Chat-V2.0-8B, should be evaluated.

**Questions:**

1.	Typos: In the caption of Figure 1, The spelling of " llustration" is wrong.

Please address the weaknesses.

---

> ### Author Response · Authors · 2024-11-22
>
> Dear Reviewer EeP4，
>
> Thank you for your insightful feedback!  Below, we address your concerns and questions individually:
> # 1. About tasks and data
> The issue of overlap among task types presents a significant concern. We generate task types for both primary and subordinate categories using prompts specifically designed to avoid overlap, thereby allowing GPT-4o to extend or create distinct task types. For instance, a prompt might be, “Please expand and supplement with new primary-level categories that do not belong to the aforementioned categories” (refer to Table 1 in the Appendix for additional prompts that yield non-overlapping task types). Ultimately, the quality of the generated task types hinges on GPT-4o. Our objective is to produce as many distinct task types as possible without human intervention.
>
> To further illustrate the variance in similarity among the generated task types, we computed a similarity matrix for all hierarchical task types using CLIP's text encoder. We then identified the maximum similarity value between each sample and all other samples, which ranged from 29.53% to 71.85%. This indicates a relatively low level of similarity. In future work, task types exhibiting slightly higher similarity can be filtered for further optimization.
>
> The diversity of tasks and data largely depends on the hierarchical generation capabilities of GPT-4o, as well as the inclusion of an extensive range of public image data sources. The tasks addressed and a select number of data samples are detailed in Appendices A.3 and A.4, encompassing a broad spectrum of tasks within the general domain.
>
> To assess the characteristics of our dataset in relation to previous studies, we provide a comparative analysis between our dataset and previous work in the following tables:
> | Dataset        | Instances | # Tasks | # Source|
> |----------------|-----------|---------|-------------|
> | LLaVA      | 150K  | 3 | Synthetic   |
> | LAMM | 196K | 8 | Synthetic   |
> | VL-Qwen   | 350K  | Unknown  |   Synthetic |
> | M3IT  | 2.4M| 40  | Private  |
> | mPlug-Owl | 150K     | 3   | Synthetic   |
> | Shikra  | 156K | 4 | Synthetic   |
> | SVIT| 4.2M| 4 | Synthetic   |
> | MultiInstruct  | 510K | 62 | Synthetic |
> | VISION-FLAN    | 1.6M  | 196  | Public|
> | TaskGalaxy | 431K |  19227 | Public+Synthetic|
>
>
> # 4. About more advanced models:
>
> For more advanced models, since LLaVA 1.6 has not officially released its second-stage instruction fine-tuning dataset, we utilize the InternVL-Chat-V2.0-8B model, which has made its second-stage instruction fine-tuning data publicly available. For the comparison, we randomly sample the same number of samples as TaskGalaxy from the officially disclosed instruction fine-tuning dataset.
>
> We fine-tune InternVL-Chat-V2.0-8B using both the original instruction fine-tuning dataset and the TaskGalaxy instruction fine-tuning dataset, ensuring that the number of samples for each is consistent with TaskGalaxy. The performance comparison table between the original instruction fine-tuning dataset and the TaskGalaxy instruction fine-tuning dataset on InternVL-Chat-V2.0-8B is as follows:
>
> | Benchmarks         | Baseline | TaskGalaxy |
> |---------------------|----------|------------|
> | MME                | 1536     | **1565**   |
> | MMBench            | 68.52    | **73.88**  |
> | MMBench_CN         | 66.46    | **70.79**  |
> | TQA                | 66.24    | **70.49**  |
> | SQA                | 90.58    | **92.71**  |
> | ChartQA            | **76.64** | 76.56     |
> | AI2D               | 75.88    | **76.75**  |
> | Q-Bench            | 57.79    | **59.65**  |
> | Q-Bench_CN         | 56.98    | **57.12**  |
> | HalluBench         | 57.51    | **58.99**  |
> | SEED               | 62.72    | **64.25**  |
> | MathVista          | 50.10    | **52.31**  |
> | MMMU               | 40.50    | **41.22**  |
> | POPE               | 86.30    | **86.90**  |
> | LLaVA-in-the-wild  | **63.20** | 62.85     |
> | MMVeT              | 46.17    | **48.86**  |
> | Average (w/o MME)  | 65.86    | **67.81**  |
>
> As shown in the table above, after fine-tuning the model using the TaskGalaxy dataset and the original InternVL-Chat-V2.0-8B fine-tuned dataset, TaskGalaxy outperforms the original baseline dataset on 14 out of 16 benchmarks. For the remaining two benchmarks, ChartQA and LLaVA-in-the-wild, the performance difference compared to the baseline dataset is minimal (less than 0.5). This demonstrates that TaskGalaxy's enhancement of task diversity is also effective for more advanced models, providing a significant boost in overall performance.
>
> About typo: We will correct the word 'illustration' in Figure 1 in a subsequent version.

---

> > ### Comment · Reviewer_EeP4 · 2024-11-25
> >
> > Thanks for the response. There remain the following weaknesses.
> >
> > 1. Merely relying on GPT-4o to ensure the task diversity is neither reliable nor convincing.
> > 2. The similarity scores between the generated tasks cannot prove any calim about the diversity.
> > 3. The improvements over LLaVA-665K is not as big as expected, since it was proposed one year ago. More state-of-the-art finetuning data should be compared.

---

> ### Author Response · Authors · 2024-11-22
>
> # 2. About baseline + TaskGalaxy data:
>
> The performance improvement observed when combining the original instruction fine-tuning dataset with our TaskGalaxy dataset can be attributed to two key variables: data quantity and task diversity. Using the baseline as a benchmark, we analyze these two variables in the ablation experiments presented in Section 3.3 of the main text ([Fig. 6]).
> To evaluate the impact of data quantity, we fixed the total number of tasks and varied the maximum number of samples per task type. This adjustment allowed us to observe the effect of data quantity on the model's average accuracy, as shown on the left side of Fig. 6. The results demonstrate a clear trend: increasing data quantity improves performance. However, as our primary focus is on task diversity, we further explore its effect in the second analysis.
>
> For task diversity, we fixed the total data quantity of TaskGalaxy and randomly sampled varying numbers of task types. The results, depicted on the right side of Fig. 6, reveal that including more diverse task types significantly enhances model performance for a fixed total data amount. This finding aligns with our expectations. Similar trends are observed in MME performance results in [Fig. 7].
>
> To further demonstrate the impact of TaskGalaxy's task diversity on model performance, we compare the baseline fine-tuning data with TaskGalaxy fine-tuning data. The comparison results, summarized in the following table, clearly show that TaskGalaxy's task diversity compensates for the limitations of existing baseline fine-tuning datasets, improving both model performance and stability.
>
> | **Benchmarks**        | **LLaVA-V1.5-7B**    |      | **InternVL-Chat-V1.0-7B**  |    |
> |-----------------------|---------------------------|-----------------------|-------------------------------|-----------------------|
> |      | **LLaVA-665K**   | **TaskGalaxy**        | **LLaVA-665K**     | **TaskGalaxy**  |
> | **MME**              |        1476    | **1520**       |   1488    |   **1512**   |
> | **MMBench**        |       63.29    |  **66.62**   |  64.86 | **65.03**      |
> | **MMBench_CN**       |  56.45     | **59.43**      |   56.41 |        **57.91**      |
> | **TQA** |   57.59 |        **58.08**      |  55.29   |   **56.15**      |
> | **SQA** |  68.77   |  **71.06**      |  65.63    | **68.88**      |
> | **ChartQA**   |  14.40    |   **19.90**      |  14.12  | **15.16**      |
> | **AI2D**    |25.29  |  **32.70**      |   35.92   | **37.69**      |
> | **Q-Bench**|       24.89      |  **30.24** |    42.89   |   **48.21**      |
> | **Q-Bench_CN**  |       31.26 |  **34.01**  |   43.73   |   **46.32**      |
> | **HalluBench**  |       47.95|        **50.95**      |    51.94   |  **53.00**   |
> | **SEED**   |       58.62  |   **59.32**      |   59.06   |    **60.44**      |
> | **MathVista**        |       28.20  | **29.30**      |   27.00  |        **30.10**      |
> | **MMMU**|  19.70   |        **20.70**      | 26.90     |    **32.80**   |
> | **POPE** |       86.30   |        **86.40**      |   86.03 |  **86.23**      |
> | **LLaVA-in-the-wild**|       47.70    |        **52.30**      | 48.30  |  **52.30** |
> | **MMVeT**    |       24.70   |   **28.60**      |   25.70   |**30.10**  |
> | **Average (w/o MME)**|       43.62  | **46.49**  |   47.17  |  **49.63**      |
>
>
> # 3. About Match and Filter by CLIP:
> Our motivation for employing the CLIP model stems from its proficiency in graphic matching, which enables us to conduct an initial screening to identify a broad range of relevant task candidate sequences—up to 30 for each image. In our experiments, we observed that the candidate task lists generally exhibited a good match with the images. However, as reviewers have noted, there is a potential for some abstract concepts not aligning perfectly with the images, or that CLIP might not be the optimal tool for this purpose. Consequently, we implemented a secondary screening phase. This involves using the existing images and their corresponding task candidate sequences to prompt GPT-4o in designing a screening mechanism, instructing the model to ascertain which task types align more closely with the image content and are more suitable for generating relevant Q&A pairs. This secondary filtering process often results in highly compatible task types; an illustrative example is provided in Figure 1 in the Appendix.
>
> The rationale behind utilizing CLIP is to efficiently filter certain task types and subsequently reduce the input length required for GPT-4o. Since it is impractical to present all task types to GPT-4o for direct screening, this approach helps streamline the process.
>
> We appreciate your valuable comments. In response, we have provided a more detailed explanation of the task type generation process and validated the effectiveness of TaskGalaxy by separating it from the Baseline data and testing it with a stronger open-source model. If you have any further questions or suggestions, please feel free to reach out!

---

> ### Author Response · Authors · 2024-11-30
>
> Thank you for taking time to review our rebuttal.
>
> # About diversity：
>
> While we cannot theoretically prove that GPT-4o can always increase the task diversity based on our proposed framework, we have empirically proved that GPT-4o does help increase the task diversity. If GPT-4o did not help increase the task diversity, adding more number of task types into the TaskGalaxy would not increase the task diversity of TaskGalaxy. In this case, TaskGalaxy with more task types would not result in better performance on several downstream benchmarks. In other words, two TaskGalaxys which contain different number of task types but same number of total data will result in similar performance on multiple benchmarks. However, as Figure 6[Left] in our manuscript shows, as the number of task types generated by our pipeline increases, the performance also increases on multiple benchmarks. These results strongly support that GPT-4o dose help increase the task diversity.
>
> On the other hand, while we cannot theoretically prove that GPT-4o can reliably provide task diversity, the generated TaskGalaxy in this study did contain more diverse tasks (empirically proved as discussed above) and can be used for various downstream tasks in the SFT dataset research community. Our approach aims to minimize human intervention for generating a wide range of diverse and non-overlapping tasks. We have incorporated additional mechanisms to refine and validate task diversity. For instance, we utilize initial task seed provision, hierarchical prompt engineering, and task embedding analysis to differentiate between tasks. The ideal method would involve incorporating expert reviews to manually verify and refine task types. This would ensure a higher level of diversity, but it conflicts with our original goal of minimizing human involvement in the task generation process.
>
> Despite its limitations, on diversity ensurance we believe this approach offers a valuable perspective on achieving reasonable task diversity under constrained human intervention.
>
> About your concern about similarity scores which cannot fully substantiate claims of task diversity, we believe that similarity can still serve as an assessment indicator to evaluate the degree of overlap among tasks, reflecting the degree of diversity TO SOME EXTENT.
>
> # About more datasets:
>
> To further highlight the advantages of TaskGalaxy, we compared its performance with that of several other SFT datasets, including LLaVA-OneVision[1], ALLaVA-4V[2], and Cambrian-1[3] in the following table. For a fair comparison, we randomly sampled the same number of samples from each dataset as in TaskGalaxy for fine-tuning. The experimental results demonstrate that TaskGalaxy consistently achieves the highest performance on most of he benchmarks across multiple model architectures, validating its effectiveness.
>
> LLaVA-V1.5-7b:
>
> |Benchmarks|LLaVA-OneVision|ALLaVA-4V|Cambrian-1|TaskGalaxy|
> |----------|---------------|---------|----------|----------|
> |MME|1251|1474|1494|**1520**|
> |MMBench|59.79|60.13|61.08|**66.62**|
> |MMBench_CN|52.84|55.39|54.46|**59.43**|
> |TQA|52.99|53.77|55.17|**58.08**|
> |SQA|**73.19**|70.05|71.03|71.06|
> |ChartQA|18.72|17.00|**20.07**|19.90|
> |AI2D|27.95|23.73|29.46|**32.70**|
> |Q-Bench|24.48|23.95|26.70|**30.24**|
> |Q-Bench_CN|33.51|33.18|33.61|**34.01**|
> |HalluBench|47.74|48.73|50.78|**50.95**|
> |SEED|30.06|40.52|49.52|**59.32**|
> |MathVista|28.2|29.2|29.1|**29.3**|
> |MMMU|17.0|18.1|19.3|**20.7**|
> |POPE|83.90|84.21|85.46|**86.40**|
> |LLaVA-in-the-wild|51.2|38.0|52.0|**52.3**|
> |MMVeT|**29.6**|27.0|25.7|28.6|
> |Average(w/oMME)|41.68|41.53|44.23|**46.49**|
>
> InternVL-Chat-V1.0-7b:
>
> |Benchmarks|LLaVA-OneVision|ALLaVA-4V|Cambrian-1|TaskGalaxy|
> |----------|---------------|---------|----------|----------|
> |MME|1350|1425|1481|**1512**|
> |MMBench|61.23|62.76|60.22|**65.03**|
> |MMBench_CN|54.74|52.78|53.01|**57.91**|
> |TQA|37.74|48.04|52.61|**56.15**|
> |SQA|66.29|66.29|67.71|**68.88**|
> |ChartQA|13.76|12.99|**16.00**|15.16|
> |AI2D|22.75|28.28|36.69|**37.69**|
> |Q-Bench|40.08|42.87|48.00|**48.21**|
> |Q-Bench_CN|42.89|44.16|41.33|**46.32**|
> |HalluBench|53.39|51.41|**54.63**|53.00|
> |SEED|40.87|48.36|56.24|**60.44**|
> |MathVista|25.0|29.4|**33.0**|30.1|
> |MMMU|24.6|27.3|30.6|**32.8**|
> |POPE|67.94|84.5|84.17|**86.23**|
> |LLaVA-in-the-wild|32.5|21.5|43.7|**52.3**|
> |MMVeT|19.4|23.5|26.8|**30.1**|
> |Average(w/oMME)|40.20|42.94|46.98|**49.63**|
>
>
> Thank you again for your thoughtful comments. We sincerely appreciate the opportunity to engage in this meaningful discussion and welcome any further suggestions or feedbacks.
>
> ```Reference```
> [1] Li, Bo, et al."Llava-onevision: Easy visual task transfer." arXiv preprint arXiv:2408.03326 (2024).
>
> [2] Chen, Guiming Hardy, et al. "Allava: Harnessing gpt4v-synthesized data for a lite vision-language model." arXiv preprint arXiv:2402.11684 (2024).
>
> [3]Tong, Shengbang, et al. "Cambrian-1: A fully open, vision-centric exploration of multimodal llms." arXiv preprint arXiv:2406.16860 (2024).

---

> > ### Author Response · Authors · 2024-12-02
> >
> > Dear Reviewer EeP4，
> >
> > We sincerely appreciate the time and effort you have dedicated to reviewing our paper. In response to your insightful feedback, we have carefully addressed each of your comments and supplemented our work with additional experiments to provide a more comprehensive response.
> >
> > As the deadline for the reviewer-author discussion is approaching, we kindly ask if our responses have sufficiently addressed your questions and concerns. If there are any remaining issues or areas requiring further clarification, please let us know. We are more than willing to provide additional explanations to ensure all your concerns are thoroughly resolved.
> >
> > Thank you once again for your valuable time and thoughtful input.

---

### Official Review · Reviewer_U9mU · 2024-11-04

**Soundness:** 2
**Presentation:** 3
**Contribution:** 2
**Rating:** 5
**Confidence:** 4

**Summary:**

This paper proposed a new instruction tuning dataset generation pipeline and a new dataset consist of ~19K tasks and ~400K samples. Through a hierarchical task defination, it aims to make diverse task types and further enhance diversity of dataset. By instruction tuned on proposed TaskGalaxy dataset, MLLM show improved performance on various benchmarks.

**Strengths:**

The pipeline proposed by the paper is diverse and scalable. It aims to increase sample diversity of instruction tuning, which is an key problem in relative field. Tuning with proposed dataset brings further improvements of MLLMs.

**Weaknesses:**

1. As the task type generation is a hierarchical process, provide some examples or statistics of level 2/3 tasks can be more clear. For example, provide a list of tasks with the most/least sample, analysis of image resolutions, distribution of tasks across levels.
2. The diverse task types actually can improve image diversity. However, image sources are still similar with existing datasets, whose diversity maybe limited. Overall review of images sources, such as the proportion of samples from different data sources, is also necessary to understand the dataset.
3. I think the experiments are not convincing enough. As baseline is instruction tuning with LLaVA-665K, TaskGalaxy finetuning actually increase the amount of data, which is shown to be a key factor for model performance. Comparing with baseline LLaVA-Instruct only and TaskGalaxy only with equal amount would be better.
4. Apart from comparison with baseline, there are several high-quality instruction tuning dataset such as LLaVA-OneVision[1], ShareGPT4V[2] and etc. Meanwhile, I think works like ShareGPT4V should be included in related work. Comparison between those works are more important to show effectiveness of TaskGalaxy, performance of models tuned with the same amount of data samples from TaskGalaxy, LLaVA-665K, LLaVA-OneVision and ShareGPT4V is more convincing.
5. To comprehensively present properties of TaskGalaxy, a detailed dataset card is preferred to show more about statistics or usage. Examples can be found at [this link](https://arxiv.org/abs/2204.01075).

[1] Li, Bo, et al. "Llava-onevision: Easy visual task transfer." arXiv preprint arXiv:2408.03326 (2024).

[2] Chen, Lin, et al. "Sharegpt4v: Improving large multi-modal models with better captions." arXiv preprint arXiv:2311.12793 (2023).

**Questions:**

See in Weakness.

**Details Of Ethics Concerns:**

As a novel dataset, ethical or social impact should be declared and discussed.

---

> ### Author Response · Authors · 2024-11-22
>
> Dear Reviewer U9mU，
>
> Thank you for your insightful feedback!  Below, we address each of your concerns and questions in detail：
> # 1. About task types:
> The names of the task types for level 2/3 stratification and the corresponding data examples are partially presented in Appendices A.3 and A.4. The detailed statistics suggested by the reviewer are helpful for a deeper understanding of TaskGalaxy. As mentioned in the text, we aim to keep the sample sizes for each task as balanced as possible. We control the number of samples per task to be between 1 and 55. The table below lists tasks with the maximum sample size of 55 and the minimum sample size of 1.
> | The most samples   | The least samples   |
> |---------------------------------------------------------|--------------------------------------------------------|
> | analysis-sports events-player performance analysis      | analysis-feature extraction-entity recognition         |
> |                                                         | analysis-focus detection-intent focus detection        |
> | suggestions-outdoor sports                              | content creation-log writing-performance review        |
> | fine-grained entity recognition-food recognition        | ocr-id document ocr-address extraction                 |
> | suggestions-kitchen appliances                          | detection-environmental change detection               |
> | ocr-environmental text ocr-shop signs reading           | ...   |
> | ...      | ...   |
>
> For the issue of image resolution, we collect the image portion of each open source dataset, keeping their original image resolution, and for all models, the input image resolution is configured to 336 by 336 pixels.
>
> For the distribution of task types across layers, the ratio of task types in the first three layers is 1:2:3, corresponding to 115, 2796, and 14,370, respectively.
> # 2. Overall review of images sources
> Considering the accessibility of data sources and the task-related nature of the image data we aim to mine, we have opted for open-source image data. The approximate data sources and their corresponding sample sizes are presented in Table 1 of the main text. To provide further insight into the image data, the following table presents the statistics of the sample sizes for the different data sources collected.
>
> | **Dataset**            | **UniGeo** | **GEOS** | **CLEVR-Math** | **VQA2.0** | **DocVQA** | **Geometry3K** | **VQA-AS** | **VQA-RAD** |
> |------------------------|------------|----------|----------------|------------|------------|----------------|------------|-------------|
> | **Sample Size**        | 1507       | 64       | 590            | 2327       | 2672       | 1182           | 658        | 242         |
>
> | **Dataset**            | **GeoQA+** | **VizWiz** | **PlotQA** | **TQA** | **FigureQA** | **MapQA** | **A-OKVQA** | **PMC-VQA** |
> |------------------------|------------|-----------|-----------|--------|--------------|----------|-------------|-------------|
> | **Sample Size**        | 2146       | 749       | 612       | 1137   | 1970         | 590      | 3391        | 4208        |
>
> | **Dataset**            | **TabMWP** | **Super-CLEVR** | **IconQA** | **DVQA** | **allava_laion** | **allava_vflan** | **Visual Genome** | **wikiart_images** |
> |------------------------|------------|-----------------|------------|---------|------------------|------------------|-------------------|-------------------|
> | **Sample Size**        | 2538       | 971             | 2560       | 1660    | 145359           | 181393           | 108249            | 62                |
>
> | **Dataset**            | **ocr_vqa_images** | **coco_train2017** | **web-celeberity_images** | **web-landmark_images** | **text_vqa_train_images** | **sam_images** | **share_textvqa_images** |
> |------------------------|--------------------|-------------------|--------------------------|------------------------|--------------------------|------------|---------------------|
> | **Sample Size**        | 12614             | 118326            | 67                       | 57                     | 3485                   | 223720     | 55                  |
>
>
>
> To further expand the existing image data, our next step is to use a generative model to create corresponding images for tasks that lack image matching.
>
> # 5. About a detailed dataset card
> To provide more statistics and usage information for TaskGalaxy, we have created a dataset card for TaskGalaxy following the suggestion. The dataset card is anonymously linked at the following address:
> ***
> [TaskGalaxy Data Card](https://drive.google.com/file/d/1lyte3ZOZh-lHSWhRi8B7_6YaLqzkmoZG/view?usp=drive_link)
> ***

---

> ### Author Response · Authors · 2024-11-22
>
> # 3&4. About more convincing experiments:
> To effectively demonstrate the validity of our dataset, we use LLaVA-665K, TaskGalaxy, LLaVA-OneVision, and ShareGPT4V individually during the fine-tuning phase. To control for variables, we randomly sample data from the other three datasets to match the size of the TaskGalaxy for fine-tuning LLaVA-v1.5-7b and InternVL-chat-V1.0. The results are presented in the table below.
>
> LLaVA-v1.5-7b:
> | **Benchmarks**        | **LLaVA-665K** | **LLaVA-OneVision** | **ShareGPT-4V** | **TaskGalaxy** |
> |-----------------------|----------------|---------------------|-----------------|----------------|
> | **MME**              | 1476  | 1251| 1501   | **1520**       |
> | **MMBench**          | 63.29          | 59.79    | 65.97           | **66.62**      |
> | **MMBench_CN**       | 56.45   | 52.84 | 59.10           | **59.43**      |
> | **TQA**              | 57.59  | 52.99   | 57.56    | **58.08**      |
> | **SQA**              | 68.77   | **73.19**           | 70.60   | 71.06          |
> | **ChartQA**          | 14.40          | 18.72   | 17.84           | **19.90**      |
> | **AI2D**             | 25.29  | 27.95   | 27.08  | **32.70**      |
> | **Q-Bench**          | 24.89   | 24.48   | 26.22 | **30.24**      |
> | **Q-Bench_CN**       | 31.26  | 33.51  | 32.51   | **34.01**      |
> | **HalluBench**       | 47.95  | 47.74   | 48.79   | **50.95**      |
> | **SEED**             | 58.62   | 30.06   | 59.26  | **59.32**      |
> | **MathVista**        | 28.20  | 28.20   | 28.20   | **29.30**      |
> | **MMMU**             | 19.70 | 17.00    | 15.60  | **20.70**      |
> | **POPE**             | 86.30          | 83.90               | 86.29    | **86.40**      |
> | **LLaVA-in-the-wild**| 47.70          | 51.20  | 49.20           | **52.30**      |
> | **MMVeT**            | 24.70          | **29.60**           | 29.00           | 28.60          |
> | **Average (w/o MME)**| 43.62          | 41.68 | 44.48           | **46.49**      |
>
> InternVL-Chat-V1.0-7b:
>
> | **Benchmarks**        | **LLaVA-665K** | **LLaVA-OneVision** | **ShareGPT-4V** | **TaskGalaxy** |
> |-----------------------|----------------|---------------------|-----------------|----------------|
> | **MME**              | 1488           | 1350                | 1191            | **1512**       |
> | **MMBench**          | 64.86          | 61.23               | 47.08           | **65.03**      |
> | **MMBench_CN**       | 56.41          | 54.74               | 40.63           | **57.91**      |
> | **TQA**              | 55.29          | 37.74               | 44.21           | **56.15**      |
> | **SQA**              | 65.63          | 66.29               | 64.55           | **68.88**      |
> | **ChartQA**          | 14.12          | 13.76               | 14.52           | **15.16**      |
> | **AI2D**             | 35.92          | 22.75               | 35.59           | **37.69**      |
> | **Q-Bench**          | 42.89          | 40.08               | 46.69           | **48.21**      |
> | **Q-Bench_CN**       | 43.73          | 42.89               | 36.38           | **46.32**      |
> | **HalluBench**       | 51.94          | **53.39**           | 52.36           | 53.00          |
> | **SEED**             | 59.06          | 40.87               | 47.24           | **60.44**      |
> | **MathVista**        | 27.0           | 25.0                | 27.2            | **30.1**       |
> | **MMMU**             | 26.9           | 23.3                | 30.3            | **32.8**       |
> | **POPE**             | 86.03          | 67.94               | 82.62           | **86.23**      |
> | **LLaVA-in-the-wild**| 48.3           | 32.5                | 30.0            | **52.3**       |
> | **MMVeT**            | 25.7           | 19.4                | 17.9            | **30.1**       |
> | **Average (w/o MME)**| 47.17          | 40.10                | 42.48           | **49.63**      |
>
> For the LLaVA-v1.5-7b model, TaskGalaxy demonstrates the strongest performance on 14 out of 16 benchmarks compared to the other three instruction fine-tuning datasets. Additionally, it achieves the highest average performance across all benchmarks, excluding MME. Similar trends are observed for the InternVL-Chat-v1.0-7b model, further reinforcing these findings. The experimental results clearly demonstrate that the diversity of task types in the TaskGalaxy data generation pipeline significantly enhances the model's generalization capability.
>
> And we will add more works like ShareGPT4V in related work.
>
> We would like to thank you for your insightful questions and suggestions regarding our work. The statistical analysis of task types, data sources, and other aspects during the data generation process has been instrumental in providing a more comprehensive understanding of TaskGalaxy. Additionally, the data cards offer a clear and intuitive way to visualize TaskGalaxy. We have also conducted further evaluations to validate the effectiveness of TaskGalaxy, and we warmly welcome more discussions on this topic!

---

> ### Comment · Reviewer_U9mU · 2024-11-25
>
> Thanks for your reply and additional information.
>
> The most important concern of mine is about the improvement of TaskGalaxy and relative datasets with the same quantity. As shown by experiments provided by the authors, results show that TaskGalaxy can bring more improvements than counterparts. I think this can resolve my concern about effectiveness.
>
> However, scale and data sources of TaskGalaxy is considered to be not satisfying. With similar data amount, performance of models finetuned by llava-onevision is pretty bad, while models finetuned with original LLaVA-OneVision is much better. It shows that dataset scale is also important in MLLM finetuning. But the construction procedure of TaskGalaxy seems to be not scalable enough. For example, there are tasks having only 1 samples, indicating data samples related are hard to collect. I think TaskGalaxy can enhance diversity and quality of data sources further.
>
> Overall, I would like to improve my rating slightly as some of my concerns have been resolved.

---

> > ### Author Response · Authors · 2024-11-25
> > **Thank you for responding to our rebuttal**
> >
> > We sincerely appreciate you taking the time to review our rebuttal and your thoughtful comments and additional concerns.
> >
> >
> > As highlighted in the experimental results and as you mentioned, LLaVA-OneVision performs worse than TaskGalaxy under the same data volume. On the right side of Figure 6 in Section 3.3 of our main text, we have also verified that as the data volume in TaskGalaxy increases while keeping the number of task types constant, the average performance on the 15 benchmarks, as well as the MME performance shown in Figure 7, improves consistently. This confirms that the data volume in a supervised fine-tuning dataset is indeed a key factor influencing the performance of multimodal models, alongside data diversity and quality.
> > Our primary goal in constructing TaskGalaxy was to enhance the fine-tuning performance of MLLMs by expanding task-related samples without relying on an excessively large dataset. The experimental results clearly demonstrate that increasing the diversity of task types can effectively improve MLLM performance, even in datasets of a similar scale, approximately 400k samples.
> >
> > The scalability of the Data Generation Pipeline for TaskGalaxy is another key feature, and we believe this pipeline is relatively easy to expand. Its scalability is primarily reflected in three areas: the expansion of task types, the subsequent expansion of data sources, and the generation of additional samples. Currently, task types are constructed using our hierarchical prompts, resulting in approximately 19k task types. These task types can be further expanded and refined into more fine-grained categories.
> >
> > Based on these expanded task types, additional open-source image data can be collected and matched with the task types to generate more data samples for TaskGalaxy. Additionally, we recognize that some task types may be challenging to match with existing open-source image data. Therefore, in future work, we plan to leverage generative models, such as Stable Diffusion, to produce image data tailored to these task types. This approach will help increase the number of data samples in TaskGalaxy and supplement task types with limited image data, further enhancing the dataset's diversity and utility.
> >
> > Regarding the total amount of sample data and the issue of having only one sample for certain tasks, we acknowledge that some specific task types, such as neuroscientific content analysis-neuropharmacology impact assessment-therapeutic target identification，scene understanding-pose estimation-pose estimation for virtual try-on, currently have very few samples in the collected open-source image data. This scarcity has led to instances where only one sample exists for these task types.
> > To address this issue, as mentioned earlier, we plan to expand the existing dataset by collecting more open-source image data or leveraging generative models to create relevant content images. Additionally, in the current TaskGalaxy, we have limited the number of samples per task type to a minimum of 1 and a maximum of 55 to balance the data proportion across task types. The size of TaskGalaxy can be flexibly adjusted by changing the maximum number of samples per task type.
> > In our analysis of the dataset, task types with fewer than 5 samples constitute 13% of the total task types, while the majority of task types have more than 5 samples. Moving forward, we will take your suggestions into account and focus on collecting more new open-source image data or utilizing Diffusion models to generate images for task types with fewer or no samples. This will enable us to continually expand and enhance TaskGalaxy.
> >
> >
> > Thank you once again for your valuable feedback and thoughtful consideration. We warmly welcome any additional questions or suggestions you may have.

---

> > > ### Comment · Reviewer_U9mU · 2024-11-25
> > >
> > > As you mentioned in the first paragraph, the paper verified the trend of model performance changing with the number of samples and tasks, but the sample sizes designed in these experiments are limited to the current scale of TaskGalaxy. The data scale of LLaVA-OneVision, couldron, cambarian, etc. is in the millions, which is much larger than TaskGalaxy. I think one point that needs additional proof is whether the number of tasks can still achieve the existing effect after the number of samples is greatly increased.
> > >
> > > In addition, with the increase in the number of tasks, some tasks are currently lacking corresponding samples. When the number of tasks is further expanded, the lack of samples may make the increase in the number of tasks meaningless. Directly increasing the sample size of each task will lead to an imbalance in the number of samples between tasks. I think this situation still needs to be verified.

---

> > > > ### Author Response · Authors · 2024-11-25
> > > >
> > > > We are very grateful for your valuable comments.
> > > >
> > > > The current design of TaskGalaxy validates the model's performance in multi-task and diverse scenarios using smaller-scale data. The results demonstrate that increasing the number of tasks can effectively enhance model performance. However, TaskGalaxy has not yet reached the scale of ultra-large datasets such as LLaVA-OneVision and Cambrian. We agree that validation with larger-scale datasets is worth further exploration. In future work, we plan to leverage generative models and expand data sources to significantly increase the dataset size while ensuring a relatively balanced number of samples across task types. This will allow us to study the impact of task numbers on model performance and verify our hypotheses.
> > > >
> > > > Thank you again for your detailed review and insightful feedback on our study. We believe that through future research and dataset expansion, we will be able to more comprehensively validate the impact of task types on the performance of multimodal models in ultra-large-scale datasets. If you have additional suggestions or ideas, we would greatly welcome further discussions.

---

> > > > > ### Comment · Reviewer_U9mU · 2024-11-25
> > > > >
> > > > > I really appreciate your comments. I think motivation and target of TaskGalaxy are somehow reasonable. Actually, current large scale benchmarks are constructed by hierarchically enlarge task coverage. I think in further development of TaskGalaxy, you need to pay attention to overlapping between tuning datasets and benchmark datasets.
> > > > >
> > > > > I think involving the generated datasets is a good option, but some validation experiments in advance to visualize or prove the effectiveness of generated samples are required.
> > > > >
> > > > > Thanks for your patience and it's nice to have such a chance to discuss with you.

---

> > > > > > ### Author Response · Authors · 2024-11-25
> > > > > >
> > > > > > We appreciate your recognition of the motivation and target of TaskGalaxy. As you pointed out, we will carefully pay attention to the potential overlap between tuning datasets and benchmark datasets in future developments to ensure evaluation reliability.
> > > > > >
> > > > > > Regarding the use of generated datasets, we agree on the importance of conducting validation experiments to visualize and prove their effectiveness. This will be a focus in our future work to ensure the quality and utility of the generated samples.
> > > > > >
> > > > > > Thank you again for your thoughtful feedback and for this valuable opportunity to discuss our work.

---

### Comment · Area_Chair_2Xxc · 2024-11-24

Dear Reviewers,

This is a friendly reminder that the discussion period will end on Nov 26th (Anywhere on Earth). If you have not already, please take a careful look at the other reviews and author responses, and comment on whether your original rating stands. Thank you.

Best, AC

---

### Meta-Review · Area_Chair_2Xxc · 2024-12-20

**Metareview:**

This paper introduces TaskGalaxy, a multi-modal instruction fine-tuning dataset with 20,000 task types and 410k Q&A samples, alongside a systematic pipeline to expand task diversity and reduce human intervention in dataset creation.  The paper initially received scores of 3,3,5,6.  The mentioned strengths include rich set of task types, flexible pipeline, and good results.  The mentioned weaknesses include more details needed on task types, image sources, rejected samples, and dataset card, issues with some of the experiments, reliance on GPT-4o, and experiments with more advanced models.  After the rebuttal and discussion, the reviewers raised their scores, with a final score of 5,5,6,8.  Most of the weaknesses were addressed, but two reviewers remained slightly negative due to remaining concerns regarding quality, effectiveness, and scalability of the proposed pipeline.  After carefully reviewing the paper, rebuttal, discussion, and author messages, the AC feels that the strengths outweigh the weaknesses and recommend acceptance.

Please incorporate all of the promised revisions in the final version.

Additionally, as one of the reviewers pointed out, "As a novel dataset, ethical or social impact should be declared and discussed."  Please add an ethics statement (instructions from https://iclr.cc/Conferences/2024/AuthorGuide):
 Authors are encouraged to include a paragraph of Ethics Statement (at the end of the main text before references) to address potential concerns where appropriate, topics include, but are not limited to, studies that involve human subjects, practices to data set releases, potentially harmful insights, methodologies and applications, pontential conflicts of interest and sponsorship, discrimination/bias/fairness concerns, privacy and security issues, legal compliance, and research integrity issues (e.g., IRB, documentation, research ethics). The optional ethic statement will not count toward the page limit, but should not be more than 1 page.

**Additional Comments On Reviewer Discussion:**

The paper initially received scores of 3,3,5,6.  The mentioned strengths include rich set of task types, flexible pipeline, and good results.  The mentioned weaknesses include more details needed on task types, image sources, rejected samples, and dataset card, issues with some of the experiments, reliance on GPT-4o, and experiments with more advanced models.  After the rebuttal and discussion, the reviewers raised their scores, with a final score of 5,5,6,8.  Most of the weaknesses were addressed, but two reviewers remained slightly negative due to remaining concerns regarding quality, effectiveness, and scalability of the proposed pipeline.  After carefully reviewing the paper, rebuttal, discussion, and author messages, the AC feels that the strengths outweigh the weaknesses and recommend acceptance.

---

### Decision · Program_Chairs · 2025-01-22

Accept (Poster)